# SQM Ageing and Atmospheric Conditions: How Do They Affect the Long-Term Trend of Night Sky Brightness Measurements?

**DOI:** 10.3390/s25020516

**Published:** 2025-01-17

**Authors:** Pietro Fiorentin, Stefano Cavazzani, Andrea Bertolo, Sergio Ortolani, Renata Binotto, Ivo Saviane

**Affiliations:** 1Department of Industrial Engineering, University of Padova, 35131 Padova, Italy; 2Department of Mechanical and Industrial Engineering, Gloshaugen, Norwegian University of Science and Technology, Richard Birkelands vei 2B, 7034 Trondheim, Norway; stefano.cavazzani@unipd.it; 3Department of Physics and Astronomy, University of Padova, 35131 Padova, Italy; sergio.ortolani@unipd.it; 4INAF—Osservatorio Astronomico di Padova, 35122 Padova, Italy; 5Regional Environmental Prevention and Protection Agency of Veneto, 35121 Padova, Italy; andrea.bertolo@arpa.veneto.it (A.B.); renata.binotto@arpa.veneto.it (R.B.); 6European Southern Observatory, Santiago 7630000, Chile; isaviane@eso.org

**Keywords:** light pollution, aerosol, solar irradiance, Sky Quality Meter, photometry

## Abstract

The most widely used radiance sensor for monitoring Night Sky Brightness (NSB) is the Sky Quality Meter (SQM), making its measurement stability fundamental. A method using the Sun as a calibrator was applied to analyse the quality of the measures recorded in the Veneto Region (Italy) and at La Silla (Chile). The analysis mainly revealed a tendency toward reductions in measured NSB due to both instrument ageing and atmospheric variations. This work compared the component due to instrumental ageing with the contribution of atmospheric conditions. The spectral responsivity of two SQMs working outdoors were analysed in a laboratory after several years of operation, revealing a significant decay, but not enough to justify the measured long-term trends. The contribution of atmospheric variations was studied through the analysis of solar irradiance at the ground, considering it as an indicator of air transparency, and values of the aerosol optical depth obtained from satellite measurements. The long-term trends measured by weather stations at different altitudes and conditions indicated an increase in solar irradiance in the Italian study sites. The comparison among the daily irradiance increase, the reduction in the aerosol optical depth, and the NSB measurements highlighted a darker sky for sites contaminated by light pollution (LP) and a brighter sky for sites not affected by LP, showing a significant and predominant role of atmospheric conditions in relation to NSB change. In the most significant case, the fraction of the variation in NSB explained by AOD changes exceeded 75%.

## 1. Introduction

Currently, a continuous growth of artificial night light (ALAN) is observed. To track the night sky brightness (NSB), different approaches and instruments are used [1,2].

The most common strategy is to measure the global emission of a sky portion, usually at the zenith of the observation site, using a radiometer, which evaluates the received power, or a photometrically calibrated instrument, in relation to the human and environmental light sensitivity [2,3,4,5,6].

The instruments have different spectral bandwidths; therefore, direct comparison of results is difficult. Scientific and commercial cameras, which can be either monochromatic or multichannel RGB cameras, also provide information on the distribution of NSB; these devices include digital single-lens reflex (SLR), mirrorless, compact, and mobile phone cameras [7,8,9].

Fish-eye cameras collect images of the entire sky, providing wide-field-of-view brightness values in addition to the zenith measurement, identifying the main sources of NSB in relation to the LP sources.

The identification of polluting facilities allows the implementation of actions to reduce the upward-scattered light in accordance with recent legislative regulations [10].

One of the most widely used and relatively inexpensive radiance meters to sample the NSB at the zenith is the Sky Quality Meter (SQM). It is essentially a radiometer that makes measurements in the bandwidth between 375 nm and 625 nm. Measurements by single instruments or by the international network of SQMs highlight a long-term trend of global reduction in NSB [11,12,13,14] making knowledge on the instrumental stability fundamental; it enables variations in the SQM output due to its ageing to be separated from those really caused by changes in NSB. A main aim of this paper is to better quantify the contribution due to ageing and to compare it with that caused by changing atmospheric conditions.

One method to evaluate the long-term trend of SQM outputs is based on the evaluation of SQM changes in response to sunlight at dusk [11,12]. Under the hypothesis of constancy of the Sun’s irradiation, variations in the measured sky brightness at dusk can be caused by ageing of the instrument and changes in the atmosphere. By looking at the variations in sky brightness at dusk and compensating for them, it is possible to evaluate NSB in terms of the initial conditions of both instrument performance and the status of the atmosphere. In Fiorentin et al. [12], different sites were analysed considering both SQMs placed in a highly ALAN-contaminated region (Veneto Region, Italy) and a site without ALAN contamination (La Silla, Chile). The results show a possible decay in performance between about 30 and less than 90 mmag_SQM_ arcsec^−2^ year^−1^ for the different instruments. The variations are comparable to the growth of outdoor artificial lighting; therefore, assessing the entity of the performance decay is of fundamental importance.

In Fiorentin et al. [12], the results obtained for the several sites, which included industrialized areas and uncontaminated areas, suggest that environmental conditions, such as a different atmosphere, could have a significant influence on the instruments’ output.

Unfortunately, the analysis of the long-term trend of the twilight measurements does not separate the effect of instrumental ageing from the influences of atmospheric changes. The long-term comparative analysis of SQM ageing in relation to atmospheric conditions presented in this work classifies the importance of the two contributions.

A first approach to the identification of decay of the instrument was to analyse some components of SQMs [15]. Among them were both parts of SQMs used outdoors and spare parts, and they included windows for the watertight housing and infrared blocking filters. The windows of the watertight housing showed no significant ageing even after 9 years of exposure and use of the instrument, while the internal infrared blocking filter showed a change in its spectral transmittance at short wavelengths of about 80%.

This ageing is not sufficient to fully account for the observed long-term trend of the SQM output.

To have a more complete understanding of the real ageing of the SQMs, they were tested by measuring their spectral responsivity in laboratory conditions. Furthermore, where solar irradiance measurements were available close to the site where the SQMs were installed, the average daily value of the solar irradiance was used to track the variability of the transparency of the atmosphere. The atmospheric transparency was also evaluated in the region where the analysed SQMs were placed; it was achieved by calculating the aerosol optical depth (AOD) from satellite measurements.

## 2. Materials and Methods

Ageing of the SQMs was analysed both in the laboratory and under the night sky. For the laboratory analysis, their spectral responsivity was estimated through the measurements of coloured light from an ad hoc-built LED source. The evaluation under real conditions was carried on by comparing the aged instrument to a new one working in parallel with it. The results made it possible to clarify how much ageing affects NSB.

To understand how the atmospheric conditions influence the SQM output, the transparency of the air was evaluated. The measurements of solar irradiance were considered information for atmospheric characterization, and, where available, their possible connection to the SQM output was assessed.

Atmospheric transparency has also been described by AOD assessment from satellite measurements, although average values over a rather large area are available (100 km × 100 km). Aerosol-cloud interaction and the aerosol-atmospheric transparency relationship are important multidisciplinary topics [16,17,18,19]; therefore, the relationship between AOD and the SQM output was analysed for all the considered sites.

### 2.1. The Sky Quality Meter

The SQM is a radiometer commonly used to measure NSB [20,21]; it is composed of a photodiode with an IR blocking filter to limit its sensitivity at long wavelengths. To take a measurement, it is simply oriented towards the zenith, and then the NSB measure is collected. Its low cost and its ability to be inserted in a protective case allow the automatic models to be used to create a network to analyse the trend of NSB. Most of the analysed SQMs that were exposed to outdoor conditions were protected by the case provided by the manufacturer; only one was covered by a dome. According to the analysis presented in [15], no significant variations in the characteristics of the exposed components were found. In the case of a variation, its contribution is considered negligible in the analysis of this work. SQM output is normally presented on a logarithmic scale, i.e., in magnitudes (m). As the radiance is weighted by the specific spectral responsivity of the instrument, in the following, the symbol m_SQM_ will be used to clarify how the power of the light is weighted.

The SQM uncertainty stated by the manufacturer is 0.1 mag_SQM_ arcsec^2^, but it is not significant for the analysis presented here. The variations analysed in this work involve the stability of the instrument and the NSB phenomenon. Although their amplitudes are lower than the SQM uncertainty, they can be obtained by heavy averaging actions on large number of samples. This also reduces the uncertainty in the results.

In the analysis presented in this work, several SQMs were considered. Four new SQMs and one SQM often used as reference in outdoor comparison were considered as references for the behaviour of non-aged instruments and analysed in the laboratory.

Two aged SQMs were also analysed in laboratory conditions. One of them had operated for 6 years over the headquarters of ARPAV at Padova and was still operating there; the second was the SQM used at the Asiago Astrophysical Observatory in the Pennar area for 10 working years. Unfortunately, the aged SQMs were not qualified when new. The components used for the production remained the same; therefore, the possible variability is within the production process, which is very simple in terms of sensor and front-end conditioning. Furthermore, when new, they were compared to each other and to the portable SQM often used as a reference, showing agreement within the stated uncertainty; this suggests that they should have behaved the same as the new ones today.

Other SQMs were also studied in this work; the recordings of their outputs were considered, looking for long-term trends and correlations with quantities characterising the atmospheric conditions. These SQMs operated at the sites described in the next section.

### 2.2. Procedure for the Ageing Assessment Under the Night Sky

New SQMs were bought to become the substitutes for the oldest instruments in the SQM Veneto network; the SQMs placed at Ekar and Pennar were replaced after 10 years. Before their installations, two new instruments were placed on the roof of the headquarters of ARPA in Padova to check that they were working correctly. They were placed side by side with the 6-year-old one. That way, their simultaneous recordings allowed a comparison of their responses in the observation of the same NSB and made it possible to measure how much ageing affects NSB measurements, albeit only for the specific exposure available at Padova, particularly in terms of the spectral distribution of the light coming from the sky. However, this comparison enabled an absolute determination of how much ageing alone affects the SQM output, as the instruments were stressed in the same way and under the same conditions.

### 2.3. The Laboratory Test Rig for the SQM Spectral Analysis

To analyse the spectral responsivity of the SQMs, they were exposed to known radiation, the distribution of which covered different wavelength widths. Figure 1 presents a sketch of the test rig used for this activity and summarises the sequence of the measuring procedure. For each one of the nine LEDs, the procedure begins with the switching on of the LED source by commanding the I/O board on the sphere using the devoted software. After few minutes, which is enough time to allow the stabilization of the LED output, the spectral radiance of the baffle is measured by the CS-1000 spectrometer (Konika Minolta (Chiyoda, Tokyo, Japan)); the spectrum is then acquired by a PC. The spectrometer is aligned to the axis of the source, leaving enough room to place the SQM close to the output port of the sphere, as shown in the figure. In this way, the baffle covers the entire field of view of the SQM. Using the software provided by the manufacturer of the SQM, its output is recorded. When enough time has passed, about five seconds, the SQM output is stabilized.

The reference spectrometer is the Konica Minolta CS-1000a; it has a range between 380 nm and 780 nm with a spectral resolution of 1 nm and an FWHM of 5 nm. Its uncertainty is 2% of the reading plus 1 digit on the luminance measurement, weighting the spectral radiance of illuminant A with the photopic sensitivity, in the range 0.01–80,000 cd m^−2^. In the confidence interval of 1 standard deviation (SD), the repeatability is 0.1% with significant spectral flatness measured within 5% of the entire operating range.

### 2.4. The Reference Laboratory Source

The 9-LED source used to evaluate the spectral response of the SQMs is shown in Figure 2.

The white reflective inner spherical surface of radius 100 mm of the sphere used as a source had a reflectivity of 0.8. The two opposite circular openings were used as the output port of radius 16 mm and as housing for 9 coloured LEDs. The baffle, which corresponded to approximately 40% of the median section of the sphere, separated the two ports. The LED light, reflected at least once, did not pass directly from the output port, improving the uniformity of the emitted radiance. The LED switching on or off was controlled remotely via PC using 9 of the 12 digital channels. Figure 2 presents the spectral radiance at the output port when the 9 LEDs were switched on separately by spacing the LEDs’ peaks uniformly across the visible range (360–720 nm) with a step of 45 nm and FWHM between 25 and 40 nm.

The different peak values had a ratio of about 5 between the strongest and the weakest LED, limiting the LED number in relation to the trade-off between the desired wavelength resolution in the knowledge of the instrument’s responsivities and the dimensions of the integrating sphere and its input port. The peak wavelengths used 1 LED, except the weakest emission at 545 nm, which required 2-LEDs. The dispersion of LED power was compensated by modifying the exposure time of the SQM and spectroradiometer. Thermal stability of the source was achieved by providing thermal transients with enough time to be negligible. The spectral radiance measured at the output port for each LED before and after each SQM/camera measurement evaluated the stability of the source by limiting the peak variation to 0.1% (negligible value in the uncertainty budget evaluation). The spectral emission of the XSL-360-SE LED, presenting a peak outside the visible range at 360 nm, induced the weakest response of the tested instruments and its radiance was therefore used to assess the lower limit of the SQM.

### 2.5. The Effect of Variations in Outdoor Lighting

SQM output variations are caused simultaneously by changes in the instrument performance and by the variations in the light from the sky, due, for example, to outdoor lighting upgrades and the installation of new lighting systems or to changes in the atmosphere.

Knowledge of the spectral responsivity of new and aged SQMs were combined with different spectral distributions of the light coming from the night sky. This made it possible to separate the output variation due to ageing from that which was caused by changes in polluting outdoor lighting due to the use of new emerging light source technologies.

### 2.6. The Analysed Sites

The instruments whose ageing was analysed were installed at sites with different characteristics in terms of light and air pollution and of altitude.

Three test sites are in Italy, in the Veneto Region. One is in Padova within the Po Valley, the most industrialised area in Italy; the site is close to sea level. The second location is the Asiago Plateau, which faces the Po Valley. Here, two SQMs are present. One close to the town of Asiago, at the Asiago Astrophysical Observatory of the University of Padova in the Pennar area, at 1050 m above sea level. A second one is at Cima Ekar Observing Station of the National Institute for Astrophysics (INAF) and is operated by the University of Padova; it is 1366 m above sea level. The third site is Passo Valles, an alpine pass in the Dolomites, located at an altitude of 2032 m above sea level and about one hundred kilometres away from the Po Valley; it is therefore partially protected from the light pollution coming from the plain.

The last instrument is located at a very clean site, in terms of both light and air pollution. It is placed at the La Silla Observatory in Chile. The site is a pass near the Chilean Atacama Desert, at an altitude of 2400 m above sea level; it is about 150 km away from the nearest important city, La Serena, which has about 120 thousand inhabitants.

### 2.7. Solar Irradiance

Atmospheric conditions could significantly affect the downward scattering of the light coming from outdoor lighting systems, and one of the simpler ways to characterise them is to analyse the variations in the atmosphere’s transparency by measuring its transmittance *T*.

In clear sky conditions, when there are no clouds, it depends mainly on aerosols. On the other side, the downward diffusion of upward-directed artificial light depends on the presence and kinds of aerosols [22,23,24,25].

A decrease in the aerosol content in the medium–low atmosphere layer could explain variation in NSB: indications from measurement show stability or darkening of the night sky observed by many low–medium-altitude SQM stations [12].

Supposing the variation in the Sun’s emission outside the atmosphere (*E_out_*) is negligible for checking the variation of interest to us, the irradiance at the ground (*E_gnd_*) is proportional to transmittance:$$E_{gnd}=TE_{out}$$

If atmospheric variations affect NSB, there should be a link between daily solar irradiance and NSB [26]; we looked for such a link.

The analysis considered data provided by outdoor sensors, which measure the sum of the contributions of direct radiation from the Sun and that diffused and reflected from the sky and cloud systems.

These sensors are components of some of the ARPAV meteorological stations in different locations in the Veneto Region. They are positioned at altitudes of up to 2200 m above sea level. Figure 3 shows the locations of all stations and highlights the sites of those are considered for the analysis of the air near the SQM sites; the first are indicated by magenta dot, the latter by red dots.

The closest station to Padova that provides recording of the solar irradiance is in Legnaro; an instrument for irradiance measurement is present in the station of Passo Valles, where the SQM is installed; a further instrument is located in the town of Asiago, near the two analysed SQMs located on the Asiago Plateau. Unfortunately, the irradiance data at Asiago are heavily affected by cloud cover; therefore, they do not represent the status of air transparency in terms of aerosol presence. In this regard, it is worth remembering that only NSB values corresponding to clear nights are considered usable to assess the light pollution status of a site. To overcome the problem of cloud cover, the data at Montecchio Precalcino, which is the closest station to the Asiago Plateau apart from Asiago itself, were associated with the SQM measurement at Asiago-Ekar analysed in this work. Obviously, it was performed only for days with clear night skies, useful for photometric and spectrometric qualification of the sky. Unfortunately, Asiago happened to have cloudy conditions during the daytime on those days, despite good conditions at night. To have indications on the air transparency, due to aerosol, the solar irradiance measurements at Montecchio Precalcino were used for those days.

The available data refer to the daily average values of the solar irradiance. The geographic coordinates of the sites are presented in Table 1.

The instrument shown in Figure 4 consists of three main parts: a transducer made of 72 nickel–chromium thermocouples, a glass protection dome, and a support body. The transducer is fixed to a metal support body and is covered by a glass dome that protects the sensor from atmospheric agents. The glass with which it is made has a high spectral transmissivity so as to be practically completely transparent to radiation in the range of interest, from 0.30 to 3.00 μm. The sensor working range is up to 1500 W m^−2^; its linearity error is less than 0.5% in the range 0.5 e 1330 W m^−2^; and its stability, which refers to its ability to maintain its metrological characteristics over time, is better than 1% per year.

### 2.8. AOD Measurement from Satellite Measurements

In conformity to [27], in this work, AOD is defined according to the equation$$AOD\left(\varepsilon \right)={-log}_{10}\left(\frac{\Phi_{e}^{i}}{\Phi_{e}^{t}}\right)$$
where $\Phi_{e}^{i}$ is the radiant flux of a collimated beam entering the upper limit layers of the atmosphere at an angle ε to the vertical and $\Phi_{e}^{t}$ is the transmitted radiant flux of that beam reaching the surface of the Earth. Therefore, AOD ranges from 0 upward.

The values of AOD used in this work are obtained from satellite observations. The satellites are Aqua and Terra; they collect information on the Earth’s water cycle and measurements of Earth’s atmosphere, land, and water [28]. They orbit at an altitude of about 700 km. The AOD values here used are obtained by processing satellite data using the Giovanni Interactive Visualization and Analysis Website [29]. The spatial resolution of the AOD data here analysed cover an area of 100 km × 100 km; this resolution allows us to have better time coverage. The area where the average value of AOD is evaluated is wide enough that only one value of AOD is available for all the sites of the Veneto Region.

### 2.9. Data Processing

All data were processed in the Matlab^®^ environment R2019b. To highlight very long-term variations, day-by-day fluctuations and seasonal changes have to be dampened in the values of SQM measurements, solar irradiance, and AOD. For this purpose, data were heavily averaged: when considering the Italian sites, each new average value was calculated using a 2-year sliding window; for the data from La Silla, a window spanning 4 months was used. Regression analysis was then applied to the filtered data.

## 3. Results

This section is divided into three main parts.

The first tries to highlight the variations in the characteristics and the performances of the SQMs that could explain the variation in their outputs. It was carried out by comparing new and old/aged instruments, both directly under the light coming from the sky when the instruments were exposed to well-controlled light sources in laboratory conditions.

The second part considers solar irradiance measurements and their relationships with AOD and with sky brightness at twilight. For the analysed sites, the values of the solar irradiance were available only for the site in the Veneto Region; therefore, the study of the correlation was considered for them only.

The last part analyses the relationship of AOD with the SQM output under the sky, both at twilight and at night; in the last case, the monthly modal values of NSB are considered. This analysis examines both the sites in the Veneto Region and the one at La Silla; they are characterised by very different atmospheric and light pollution conditions. Unfortunately, we do not have measurements of solar irradiance at La Silla. Therefore, the analysis of the data from this site is considered separately.

### 3.1. Performance Comparison Between New and Old/Aged Instruments

As the oldest instruments in the SQM Veneto network needed to be replaced, the new ones were characterised and compared with the aged ones, both under the night sky and in the laboratory.

#### 3.1.1. Comparison Under the Night Sky

The new SQMs were placed in parallel with the one at Padova (ARPAV headquarters), which is 6 years old. It allows a first comparison of the behaviours of the new SQMs and the 6-year-old one, analysing the same NSB for 27 (from 24 January 2022 to 20 February 2022) and 32 nights (from 21 February 2022 to 25 March 2022), respectively, for the two new SQMs; each period was approximately one full lunar cycle long.

The magnitude of the NSB was measured by the three SQMs, and the difference between each new instrument and the old one is considered herein. Figure 5 shows the histograms describing the estimates of the probability density of appearance for values of the difference. The two distributions are similar: they present an average equal to 0.078 mag_SQM_ arcsec^−2^ and 0.077 mag_SQM_ arcsec^−2^ and a standard deviation equal to 0.085 mag_SQM_ arcsec^−2^ and 0.055 mag_SQM_ arcsec^−2^, respectively. The dispersion is certainly due to the difference among the SQMs but also to their different positioning on the roof of the headquarters of ARPA of the Veneto Region. Furthermore, the values of the standard deviation are close to the uncertainty of the SQM declared by the manufacturer. It is not possible to make a comparison between the recordings of the two new SQMs, since they were collected separately at different times. The output difference between the new instruments and the old one of about 0.08 mag_SQM_ arcsec^−2^ can be attributed to the variation in the responsivity of the old one, but it also depends on the spectral distribution of the light coming from the night sky during the recording periods. Since the recording time intervals are consecutive and of limited duration, it can be supposed that the spectral distribution of the light downward from the night sky is the same, which makes it feasible to compare the values of the two differences and to assume one average difference value of NSB measured by the new instruments and the older one, equal to about 0.078 mag_SQM_ arcsec^−2^.

#### 3.1.2. SQM Responses to the LEDs

Procedure for the ageing assessment in laboratory under the LED sources considers together the four new SQMs and the one SQM often used as reference in outdoor comparison. All of them are considered not affected by ageing, and the average behaviour of these 5 instruments is the starting point of this analysis. The other two SQMs used at Padova and Asiago-Pennar were also lit by the LED source, and their outputs were studied to highlight their ageing. Figure 6 shows the ratio of the linear output of each SQM normalised to the total radiance at the output port of the LED source. The values O_l,n_ are obtained by linearizing the logarithmic output of SQMs divided by the radiance of the source output port according to the expression$$O_{l,n}=\frac{10^{\frac{m_{SQM}}{2.5}}}{R}$$
where m_SQM_ is the usual logarithmic output of the SQM, the numerator on the right-hand side of the equation is the SQM’s linearized output, and R is the radiance of the output port.

Each output is presented at the peak wavelength of the corresponding LED. The result is not the spectral responsivity of the SQMs, in fact, each value is obtained by lighting the instrument with a source with the finite bandwidth of the LED; however, it is a signature of the analysed instrument. A simple linear interpolation was used, just to make the trend visible.

The blue curve shows the average behaviour of new instruments; for each measure, the vertical bar describes the dispersion within the five instruments as standard deviation; the maximum dispersion is 4.1% of the value. The horizontal bar straddling the peak wavelength of each LED reminds that the output is obtained by using LEDs, which do not have a negligible bandwidth.

The cyan line presents the result for the SQM at Padova used for 6 years, at few meters AMSL, and the green one shows the response of the SQM used for 10 years at Asiago-Pennar at about 1000 m AMSL. For both responses, the distance from the average behaviour of the new SQMs is apparent, also considering their dispersion. For the SQM used at Padova significant variation is visible from 450 nm down, while, for the SQM from Asiago-Pennar, the difference is also evident at longer wavelengths, starting from 550 nm.

#### 3.1.3. Measurement of Spectral Responsivity

From the response of the SQMs to the LED source, the spectral responsivity of each analysed SQM was estimated according to the procedure described in [8]. As an example, the case of new SQMs is presented in Figure 7a. The continuous line represents the estimate of the spectral responsivity in arbitrary units, and the black squares represent the SQM measures normalised to the radiance of the output port of the LED source, as in Figure 6. The blue circles show the estimates of the measures the SQM should provide when lit by the source when each LED is switched on, the values are calculated from the spectral responsivity presented in the Figure 7a and the spectral radiance of the LED shown in Figure 2b. A good agreement with the real measured values is visible, confirming the good reconstructed spectral responsivity of the SQM.

The spectral responsivity of the aged instruments was also estimated according to the same procedure; they all are presented in Figure 7b. Again, the continuous blue line shows the average spectral responsivity of new SQMs; above and below it, the blue dotted lines represent the one-standard-deviation uncertainty boundary. The red line corresponds to the responsivity of the SQM used at Padova after 6 years of work, and the magenta line describes the SQM used for 10 years at Asiago-Pennar.

For the outdoor exposed SQMs, a clear decay is visible in Figure 7b at the shortest wavelengths; in particular, the SQM used at Padova presents a significant attenuation of its responsivity at wavelengths below 470 nm; while for the SQM from Asiago-Pennar, the decay also appears at the longest wavelength, and the difference compared to the reference is already evident for wavelengths longer than 550 nm.

#### 3.1.4. Effect of Spectral Decay

Figure 8a presents again the responsivities of the two SQMs at Padova and Asiago-Pennar and the average responsivity of new SQMs, they are shown together with the sky spectra recorded in locations characterised by different light pollution. The amplitudes of the spectra in the figure are not significant, while it is apparent that going from low polluted to high polluted sites, the spectrum shape changes a lot. In particular, there is an increase in the range between 550 nm and 600 nm, and around 460 nm, which is now caused by the introduction of the white LEDs for outdoor lighting.

The spectral responsivity of the SQM weights the downward-directed light from the sky in different ways at different wavelengths; the weight depends on the decay of the SQM. It is interesting to analyse how much the combination of the changes in the sky spectra and the SQM responsivity can produce a variation in the SQM output. The results are presented in Figure 8b for the two SQMs of Padova and Asiago-Pennar, considering as reference the responses of a new SQM. The outputs of the three SQMs were evaluated for every spectral distribution, then the differences were calculated by subtracting the value corresponding to the new SQM, this for each light distribution. Therefore, in Figure 8b the zero value corresponds to a new instrument, while the circles at 6 years correspond to the discrepancies that could be shown by the aged SQM at Padova, and the circles at 9 years show the possible spread of the output of the aged SQM at Asiago-Pennar. Each circle in section (b) corresponds to the spectral distribution in section (a) with the same colour.

After 6 years, the output of the SQM at Padova could vary from a minimum of 0.01 mag_SQM_ arcsec^−2^ to a maximum of about 0.074 mag_SQM_ arcsec^−2^. Larger variation in the output of the SQM at Asiago-Pennar could be expected after 9 year of exposure, from 0.24 mag_SQM_ arcsec^−2^ to 0.36 mag_SQM_ arcsec^−2^. From these values, an average slope of about 5.3 mmag_SQM_ arcsec^−2^ y^−1^ and 31 mmag_SQM_ arcsec^−2^ y^−1^ in the decay of the SQM output can be estimated for Padova and Asiago-Pennar, respectively. This amounts partially justifies the slope of the long-term trend of the sky brightness at twilight detected for the SQM outputs of 53 mmag_SQM_ arcsec^−2^ y^−1^ at Padova and 85 mmag_SQM_ arcsec^−2^ y^−1^ at the Asiago-Pennar site as presented in [12].

The possible maximum variation in the Padova SQM corresponds to the most recent spectral recording and to a significantly polluted site. This spectrum is represented by the top curve among the spectral distributions in Figure 8a, and it is very close to the sky spectrum present during the test of the new SQM in Padova. As a consequence, the possible maximum variation due to ageing presented in Figure 8b for the SQM at Padova is very near to the value 0.078 mag_SQM_ arcsec^−2^, which is the average of the difference between new SQMs and the 6-year-old SQM at the Padova site, found during the direct comparison under the sky at Padova (see Figure 5).

### 3.2. Solar Irradiance at Ground Level

Available data provide the average daily solar irradiance; therefore, they describe the average behaviour during the day of the atmosphere at the considered site. Figure 9 shows the evolution of the daily solar irradiance measured at sites close to the locations where the analysed SQMs are installed. The data refer to the recordings of Legnaro-Padova, to the instrument at the station of Passo Valles, and to Montecchio Precalcino-Asiago-Ekar.

For all three stations, a high variability of the data is evident throughout the year: there are seasonal variations, characterised by the slow varying “sinusoidal” component, and cloud cover changes, that mainly cause the fast variations in the recording.

None of these two types of variations in the average daily solar irradiance are of our interest, as we are looking for the long-term trend of NBS and, therefore, searching very slow and long-term variations in the atmospheric conditions. Consequently, solar irradiance data were heavily averaged, each average new value is calculated using a 2-year sliding window. The results of this operation are presented in Figure 9 by the red lines. The amplitudes of the variations in the mean tendencies are very small compared to the raw daily values, to highlight these averaged quantities a separate plot is needed, it is presented in Figure 10. The results for Legnaro and Montecchio Precalcino indicate a decadal increase in the solar irradiance starting from the year 2014 of about 2.5 W m^−2^ y^−1^. The filtered data from Passo Valles have a less clear tendency, showing an initial decrease and a subsequent increase of the solar average irradiance from the year 2017. The solar irradiance can be linked to air transparency, and an increase in irradiance can represent an improvement in transparency. The change could be caused by the updating of many heating systems in the Po Valley, switching from the use of diesel to methane.

In addition to the data already presented in Figure 9, Figure 10 also shows the average of the solar irradiance at Asiago; it presents significant oscillations even at such low frequency; they are the effect of the presence of clouds, and prevent the use of this quantity to represent the presence of aerosols in the atmosphere.

#### 3.2.1. Daily Irradiance and AOD

AOD is a quantity describing transparency of the air more directly, although, in our case, its measurements are available only over a large area; therefore, solar irradiance is also used for the same purpose with data from sites closer to the SQM stations [30,31,32].

To analyse the validity of this choice, the relationship between AOD and solar irradiance is studied looking at the correlation between these two quantities referred to the site of Padova, for example. Figure 11a shows the time evolution of the average daily values of the solar irradiance and the daily values of AOD from satellite measurements. The variability of both data sets is evident; the sinusoidal slow oscillations are caused by changing season, the fastest day to day apparently random variations are imputable to various causes, such as cloud cover changes, variations in the aerosol content or of air pollution. In the same image the two lines show the slow tendency of the two quantities, the values are obtained by heavy filtering the daily data by a moving average with a 2-year sliding window. To better examine the long-term behaviours, the section (b) of the same figure shows a zoom on the two curves. Over time, a clear trend towards lower values of AOD and higher values of solar irradiance can be observed. The values of the slopes of a linear regression line fitting of the average irradiance data are about 2.84 ± 0.026 W m^−2^ y^−1^ for Padova and 2.52 ± 0.036 W m^−2^ y^−1^ for Asiago-Ekar. The trends of the two quantities suggest that they present a negative relationship, the analysis of the correlation between them was performed to evaluate a possible link between them. Figure 12 presents by blue circles the average data of AOD versus the averaged solar irradiance values, the data are reduced considering only one value per month; it is the representative of all data of the month; on the other side, analysing all daily average values only apparently increases the information on the trend of both quantities, since a heavy average action has been made. The Pearson correlation coefficient between the averaged values of the two quantities is high, equal to 0.88. The result of the regression analysis is represented by the red continuous line, and the dashed lines astride show the boundary with 95% uncertainty, and described in Table 2 in terms of coefficients of the linear regression model.

#### 3.2.2. Daily Irradiance and Twilight Calibration

Looking at the mean daily solar irradiance of Figure 9, it is apparent how much it depends on the air quality. The same also happens for the SQM measures at twilight [12], which are used to calibrate the SQM and sometimes to assess the decay of their performances and ageing. The correction is the difference between the current measure of the sky brightness at dusk and the value of the same measure at the beginning of the recording. Here, the possible link between the daily irradiance and the brightness of the sky at dusk is evaluated. As above, the average trend of both quantities is considered by applying a heavy moving average with a 2-year sliding window to compensate seasonal and random causes of variability. The results of the analysis are presented in Figure 13 for the three Italian analysed sites. A significant correlation is present for the sites of Padova (ρ = 0.62) and Asiago-Ekar (ρ = 0.75), while for Passo Valles the correlation coefficient has a very low value (ρ = 0.15). In particular, for the sites of Padova and Ekar, at the lowest values, the variations in the sky brightness at twilight do not appear evident in relation to the solar irradiance, while it becomes clearer for values of irradiance and SQM output. The low values of correlation coefficient can be attributed to a different geometry in the propagation of the light during the day, when the Sun is above the horizon, and at dusk, when the Sun is in the range from −7° to −6° below the horizon. A possible cause of the lack of correlation could be the limitation of not having the two measurements at the same site, despite Passo Valles having both measurements at the same place and presenting the lowest correlation.

A linear trend is considered with its 95% uncertainty boundary to describe the tendency, even if the laws of the trends do not appear completely clear. The values of the parameters of the regression lines are presented from Table 3, Table 4 and Table 5. However, for the first two sites a tendency towards higher sky brightness at dusk is evident, corresponding to a solar irradiance increase. Consequently, the correction based on the twilight method seems to depend on air transparency, which is evaluated here by daily solar irradiance. Therefore, the twilight calibration cannot provide unambiguous information on SQM degradation, as was stated above.

The regression lines have slopes of about 5 mmag_SQM_ arcsec^−2^ W^−1^ m^2^ for Padova and 10 mmag_SQM_ arcsec^−2^ W^−1^ m^2^ for the Asiago-Ekar SQM. Combining these results with the increasing slope of the solar irradiance of about 2.5 W m^−2^ y^−1^, the variation along time of the solar irradiance can justify a trend towards an increase of the sky brightness at twilight which has an average slope of 14 mmag_SQM_ arcsec^−2^ y^−1^ for the SQM at Padova and of 25 mmag_SQM_ arcsec^−2^ y^−1^ for Asiago-Ekar. These values have to be compared with the trends presented in [12] for the sky brightness at twilight of 53 mmag_SQM_ arcsec^−2^ y^−1^ for both sites. The contribution to the trend due to the increasing of the transparency of the air, here described by the variation in the solar irradiance, appears important and is about one-third or one-half of the tendency observed by looking at the sky brightness at twilight, for the two Italian sites.

#### 3.2.3. Daily Irradiance and NSB

For clarity, Figure 14a presents again the solar irradiance averaged over a 2-year sliding window at Legnaro, Montecchio Precalcino, Passo Valles, already presented in Figure 10. Here, the sets of values are limited to the duration of the SQM recordings in the related sites shown in the legend. Aside, section (b) shows by circles the values of the monthly modal values of NSB at Asiago-Ekar, as example. The same section of the figure highlights the average trend of the modal values by the red continuous line, a 2-year sliding window is used for the moving average. The comparison of the trends of the average solar irradiance and of the average NSB makes it possible to perceive a possible correlation between the two quantities.

The monthly modal values of NSB were plotted versus the average solar irradiance and represented by circles; the plots in Figure 15a corresponds to the site of Padova, while the section (b) of the same figure shows the situations at Asiago-Ekar and Passo Valles. The red continuous lines show the average trends, considering the average values obtained by applying a moving average with a 2-year window. Superposed on all, the blue curves show the behaviour of the functions considered for a fitting of the monthly data. For Padova and Passo Valles, the fitting function is linear, while the data of Asiago-Ekar are fitted by an exponential function, which better follows the average trend; the parameters of the fitting functions are presented from Table 6, Table 7 and Table 8.

The modal values of NSB at Padova, which is within the Padana Plain, show an increase as irradiance at ground level increases. The link between the two quantities seem to be significant, the Pearson correlation coefficient is equal to 0.72. Higher values of the solar irradiance mean more transparent atmosphere, in particular a reduction of the effect of the main aerosol layer, which scatters downward less artificial light coming from ground.

The modal values of NSB at the Asiago-Ekar station, which is on the Asiago Plateau close to the Padana Plain, show an increase as irradiance at ground level increases, with a correlation coefficient equal to 0.83. The phenomenon seems to be stronger than at Padova, for the recorded irradiance values, and the modal values of NSB increase more as the irradiance and the air transparency improve.

NSB values at Passo Valles, which is far from strong light pollution sources, seem to be weakly linked to daily irradiance. They present a slight decreasing with the improvement of air transparency. A possible explanation proposed by the authors is that the SQM is more influenced by the upper aerosol layer that blocks the natural radiation of the night sky, when air transparency increases, this light can reach the instrument more freely.

From the results relating to Asiago-Ekar and those relating to Passo Valles, it seems the modal value of NSB of both sites can reach the value of about 21.1 mag_SQM_ arcsec^−2^ for very transparent air, which can be associated with the daily average solar irradiance of about 320 W^2^ m^−2^. In this analysis, higher values of the Pearson correlation coefficient are found than when analysing the brightness at dusk; the result is mainly true for Passo Valles. It can be due to more similar geometry in the present case.

### 3.3. AOD Effect on Measurement of Sky Brightness

The results are presented separately for the analysis in the Veneto Region and at La Silla, the last site is influenced by far light polluting sources, and is almost isolated on the saddle, furthermore due to its peculiar positioning presents atmospheric and light pollution conditions very different from the other sites.

#### 3.3.1. AOD and Brightness at Twilight in the Veneto Region

As previously stated, only one evaluation of AOD is used for all sites in the Veneto Region, owing to the reduced resolution of the satellites. Figure 16 presents the relationships between the AOD values and the values of the sky brightness at twilight measured by the SQM in the three sites in the Veneto Region. A significant correlation was found for the site of Asiago-Ekar and Padova, with a Pearson correlation coefficient of 0.82 and 0.58, for the two sites respectively. For all sites, a linear regression is also applied to the data (blue circles), it is represented by the continuous red line, the dashed lines show the boundary of the regression with a 95% confidence level.

For the site of Passo Valles, the Pearson correlation is zero and the linear regression model completely fails. The difference is probably due to the different position of the three sites relative to the main aerosol layer. AOD and NSB are closely related to altitude [33]. The site of Padova at the lower altitude is certainly below the main layer; the same is probably true also for Asiago-Ekar. Therefore, their data are more influenced by the variations in the air transparency, here expressed by AOD. On the contrary, the light coming from the sky more directly reaches a site above the main aerosol layer, as it may be at Passo Valles, causing a higher average measured sky brightness at twilight visible in Figure 16c; in fact, the Sun’s light is scattered downward by the upper aerosol layers and is not attenuated by the lower, more important one. The measurements of the SQM at Passo Valles seems to be not affected significantly by the variations in the evaluated transparency of the air. Here we must also remember that AOD is evaluated as average values over almost the whole Veneto Region, therefore, including low altitude layers, which obviously do not influence in the same way the transmission of the Sun’s light at twilight towards sites near sea level or sites at about two thousand meters above sea level.

Overall, the results show that variations on the SQM output highlighted by the twilight method could depend on the air transparency, here described by AOD; furthermore, its influence varies depending on the site, ranging from almost zero to values well above the uncertainty declared by the SQM manufacturer.

Details on the coefficients of correlation and of the parameters of the regression lines are presented in Table 9, Table 10 and Table 11; the uncertainty of these parameters is presented here as well, described by one standard deviation.

#### 3.3.2. AOD and Modal Value of NSB in the Veneto Region

Figure 17 shows the relationship between AOD and the modal values of the night sky brightness for the site of Padova (a), and the sites of Ekar and Passo Valles (b). The values of both AOD and the NSB mode are obtained from a moving average with a 2-year sliding window. Monthly data are represented by the blue circles, their linear regression is depicted by the continuous red line, the dashed lines indicate the boundary of the regression with a 95% confidence level. Table 12, Table 13 and Table 14 present the values of the Pearson regression coefficient values and the regression model parameters of the three sites, respectively.

The situation at Padova is the simplest, increasing values of AOD mean the more important is becoming the main aerosol layer, which downward scatters the artificial polluting light towards the SQM. A strong link is present between AOD and the modal values of NSB, which is highlighted by the Pearson correlation coefficient equal to −0.82.

Passo Valles is partially protected by the scattering of the light coming from the Po Valley by the mountains around the pass itself. Furthermore, it is located at an altitude that is higher than the main aerosol layer. The most important variations in the considered values of AOD are probably due to changes in the lower and main aerosol layer, but higher values of AOD also correspond to decreases in the transparency of the layers at higher altitude. It is their variations that influence the measurements at Passo Valles by reducing the portion of the natural light flux reaching the instrument. The situation is presented in Figure 17b by an increasing trend of the modal values of NSB at Passo Valles, as AOD increases.

The site of Asiago-Ekar is at lower altitude than Passo Valles and is probably close to the main aerosol layer, maybe below it. As for Padova, as well as for Ekar, the higher the AOD values, the more light coming from the Po Valley is scattered downward by the main aerosol layer toward the instrument. It causes the negative trend of the modal values of NSB versus AOD, which is presented in Figure 17b and Table 13. The reduction of the natural flux reaching the instrument due to the worsening of the aerosol layer at higher altitude is of secondary importance, and the effect on the scattering of artificial light dominates. Overall, at Ekar, the increasing of AOD brings to a growth of NSB similar to that at Padova, even if the correlation between the two quantities is a bit lower in the Ekar case, and the Pearson correlation coefficient is equal to 0.75.

From the results relating to Asiago-Ekar and those relating to Passo Valles, it seems the modal value of NSB of both sites can reach the value of about 21.1 mag_SQM_ arcsec^−2^ for very transparent air, which can be associated with the daily average solar irradiance of about 320 W^2^ m^−2^.

#### 3.3.3. AOD Versus Brightness at Twilight and the Modal Value of NSB at La Silla

To analyse the status of the SQM, the twilight method is considered. Figure 18 shows the daily values of the average sky brightness at sunset and dawn by blue circles. In the same figure, the yellow curve represents the values obtained by a moving average with a sliding window 4 months wide; the green curves close to it represent its one standard deviation of variability. To estimate a possible tendency in the SQM output, the linear regression of the daily values was considered (red line) with its boundary (blue curves). Figure 18 also displays the value of the slope of the regression line, the value −0.004 mag_SQM_ arcsec^−2^ y^−1^ is very small and, although it could be statistically significant, seems to show no significant variation in the behaviour of the instrument, in terms of its ageing. On the contrary, the moving average clearly shows oscillations with a period of one year, they can be associated with seasonal variations in the atmosphere. Superposed to them, other changes are present, sometimes with comparable amplitudes.

A possible relationship between the sky brightness at twilight and the transparency of the atmosphere was also searched for in the data from the SQM at La Silla. Figure 19 presents the plot of monthly average values of the sky brightness at twilight in the period from 2018 and the beginning of 2024. The straight horizontal blue line shows the result of a linear regression, performed just to try to highlight a long-term trend. The coefficients of the line are presented in Table 15, again, no apparent trend appears. The same figure also presents the plot of the monthly average values of the aerosol optical depth obtained from satellite measurements. Here and in the following, only days for which data are available for both analysed quantities, brightness and AOD, are selected. The oscillations of the AOD are clearly synchronous with the oscillation of the sky brightness and have a periodicity of one year, denoting that they are seasonal. An increase in AOD, i.e., a less transparent atmosphere, heavily filters the Sun’s light measured by the SQM at twilight and brings to higher SQM magnitude values. An anomaly seems to be present in the year 2022, when a strong variation in AOD appears, even if it does not correspond to a high variation in the SQM measurements. This possible incoherence in the relationship between the two quantities could be at least partially justified by remembering that while the sky brightness here considered is the average values at sunset and dawn, when the Sun is about at −6° below the horizon, AOD is always evaluated during the daytime, when the Sun is up, and atmospheric conditions could be different when the two kinds of data are collected.

To check for a possible trend in the AOD values, the linear regression on those values was considered. It is performed over two sets of data: the first corresponds to all data but those greater than 0.07, and the second to the whole set of recorded data. The value 0.07 is below 0.1, which is considered to indicate a crystal-clear sky with good visibility [34]. The choice of removing data with higher AOD agrees with the selection of the SQM data, which corresponds to dusk and dawn before and after clear nights suitable for spectrometric and photometric measurements on stars, therefore having very good visibility.

This way, the effect of the peak in the last year can be separated in the trend evaluation. The straight magenta line in Figure 19 shows the result of the linear regression on the first set, the dashed magenta lines show its boundary with a 95% confidence level. The regression line (see Table 16) has a slope of −0.0002 y^−1^, with an uncertainty four times this value. It means there is no detectable trend. Furthermore, the red line, representing the linear regression of the full set of data, is inside the 95% confidence level of the first regression, it suggests that the high peak does not significantly move the tendency.

To highlight a possible relationship between the sky brightness at twilight and the aerosol optical depth, as shown in Figure 20, the first was plotted as a function of the second. Then the Pearson coefficient of correlation was evaluated, and the linear regression fitting was considered. For this purpose, only the data of the days when AOD is less than 0.07 are considered, according to the evaluation presented above. The other data are represented by red circles. The continuous line represents the linear fitting, and the dashed lines show the boundary of the regression with a 95% confidence level. The parameters of the fitting line are presented in Table 17. The slope of the regression line has a relative uncertainty of about 38%, and the coefficient of correlation is just below 0.6. Both things indicate a possible weak dependence of the twilight sky brightness on the AOD, even if other quantities also influence the sky brightness, clearly causing the dispersion of the data. Furthermore, it has to be remembered that the two quantities are measured at different times: AOD is obtained during the day, while sky brightness is considered at twilight; therefore, in the meantime, significant variations in the AOD could occur. This clearly reduces the correlation between the measured quantities.

It is also interesting to analyse if and how much AOD affects the sky brightness at night. Figure 21 shows the time evolution of the monthly mode of the night sky brightness at La Silla as blue symbols. The same figure presents, as red symbols, the monthly average values of AOD, already considered. The thin blue and red lines show the trend of a moving average of the two quantities, a 4-month sliding window was used. They better highlight the seasonal component with a periodicity of one year, both on AOD and on the monthly mode values of brightness at night. Again, the oscillations on the AOD are synchronous with the oscillation of the sky brightness. To analyse the possible presence of a long-term trend in the night sky brightness, a linear regression of the data was performed. The results are summarised in Table 18 and presented in Figure 21, where the thick continuous line represent the regression line, and the dashed lines show its boundary with a 95% confidence level. The outcome is a possible growth trend of the mode values of the night sky brightness, lower magnitude values. The apparent tendency is very weak, as shown by the small brightness change in 6 years of about 0.1 mag_SQM_ arcsec^−2^ and by the value of the slope of the regression line, which is comparable with its uncertainty, of about 68%. Since no trend in the AOD was detected, indicating that atmospheric conditions had not changed, the possible variation in the night sky brightness could be due to new lighting installations, perhaps in towns hundreds of kilometres away from the La Silla Observatory and/or new lighting installations along the Panamericana.

Figure 22 shows the monthly mode values of the night sky brightness versus AOD at La Silla, and a possible link between the two quantities was analysed by considering the linear regression between them, as well as the Pearson correlation coefficient. As for the brightness at twilight, a small coefficient of correlation was found for the modal night values as well. Furthermore, the slope of the regression line had a high uncertainty of 40%, see Table 19. However, its positive value agrees with the physics of the phenomenon: an increase in AOD, i.e., a decrease in air transparency, reduces the ability of the light coming from natural sources to reach the SQM, and therefore a darker sky is observed. The value of the intercept for AOD equal to zero of 21.6 ± 0.1 mag_SQM_ arcsec^−2^ is close to the value 21.71 ± 0.24 mag arcsec^−2^ observed at the Paranal Observatory in Chile, at about the same altitude, but about seven hundred kilometres north of La Silla [35]. It confirms the result of the regression and the possible link between the two quantities.

Despite presenting different values of the monthly mode of NSB and very different values of AOD, which correspond to a cleaner atmosphere, the trend at La Silla was like the one observed for the site of Passo Valles in Italy; this occurred because, in both cases, the measurements of NSB were recorded from a site that was above the main aerosol layer.

## 4. Discussion

The result obtained for the analysed instruments, in particular at the Italian sites, can be associated with their positions relative to the aerosol layers.

The sketch in Figure 23 simply describes the scattering of the upward-directed artificial light and the downward-directed natural light for the Italian sites, with different position relative to the main light pollution sources and aerosol layers. Wider or thicker arrows represent greater luminous flux involved. From this simple description, we can argue that reducing aerosols in the lower layer causes less artificial flux to be scattered downward, reducing NSB seen from sites below the layer. Conversely, a reduction in aerosol, particularly in the upper layers, allows the natural light easier access to the instruments, which measure higher NSB.

The situation of the site of La Silla can be supposed to be similar to that of Passo Valles; in this case as well, the instrument is placed above the main aerosol layer, and an increase in the aerosol in the upper layer mainly shielded the light coming from outside the atmosphere. Therefore, the cleanest air, described by the lowest AOD values, is associated with the highest values of sky brightness, i.e., the lowest SQM measurements.

An attempt to measure the goodness of fit of the model can consider the coefficient of determination R^2^, which describes how well the regression predictions approximate the real data points. In further detail, R^2^ can be considered as the ratio of the variance explained by the model to the total variance. In our case, the closer it is to unity, the more the variations in sky brightness are explained by changes in air transparency.

The values of the coefficients of determination are presented in Table 20 for the four analysed sites and for the different analyses of the influence of air transparency, through the average daily solar irradiance and AOD, on both the sky brightness at dusk and the modal values of NSB.

The variations in the air transparency explain, on average, about 60% of the variations in both the sky brightness at dusk (B_dusk_) and the modal values of NSB (M_NSB_) for the site of Padova and about 70% for the measures at Asiago-Ekar. The lower value of R^2^ for the relationship between the sky brightness at dusk at Padova and AOD, compared to the one obtained considering the solar irradiance, does not agree completely with the strong relation between AOD and solar irradiance presented in Figure 12.

The site of Passo Valles confirms the weak or null detected sensitivity of the sky brightness at dusk to air transparency; it could be due to a lack of significant variations in this quantity at this site. It appears that changes in the air above the site do not affect the scattering of the light from the Sun below the horizon. On the contrary, variations in air transparency explain a portion of the changes of the modal values of NSB, from 19% to 35%. Air transparency plays a role in the NSB variations, although other quantities can have more significant effects, including the number and type of celestial bodies within the field of view of the SQM and variations in outdoor lighting systems.

The variations in AOD at La Silla explain about 30% of the variation in the sky brightness at dusk and at night. As at Passo Valles, other quantities play more important roles; this happens in particular for the long-term trend.

From the regression of the data of the sky brightness at twilight versus AOD, an increment in the brightness was found with a slope of −3.8 mag_SQM_ arcsec^−2^ and −7.9 mag_SQM_ arcsec^−2^ for the sites of Padova and Asiago-Ekar, respectively. The observed variation in the monthly mode of NSB at the same sites presented a slope of −2.3 mag_SQM_ arcsec^−2^ and −5.6 mag_SQM_ arcsec^−2^. The differences between the slope related to the twilight method and the slope obtained accounting for NSB were significant when compared with the uncertainty on the slopes themselves. For both sites, the correction suggested by the twilight method seems to overcompensate for the effect of the variation in air transparency.

For the site of Passo Valles, the twilight method does not suggest any need for correction, while the analysis of the dependence of the monthly mode of NSB on AOD indicates that less transparent atmosphere causes a reduction of the measured value of the sky brightness, a compensation should be consequently applied.

Considering the site of La Silla, where the atmospheric configuration is probably simpler, the slope obtained analysing the light at twilight is 8 ± 2.7 mag_SQM_ arcsec^−2^, while the slope evaluated considering the modal value of NSB is 9 ± 3.6 mag_SQM_ arcsec^−2^. In this case, the two intervals present a superposition, which could indicate the correction prosed by the twilight method effectively compensates the loss in NSB measurement. For this site, we must also consider that the variation intervals of the value of AOD, of the brightness at twilight, and of the modal values of NSB are very small, therefore they could be affected by second order changes in the environmental conditions, which are not described by AOD.

To have a possible answer to the different behaviour of the brightness at twilight and of NSB versus variations in the environmental conditions, an analysis of the possible correlation between the brightness at twilight and the modal values of NSB can be performed. If the two quantities are measured in similar conditions, their correlation should be close to unit value, and the regression line fitting the data should have a unitary slope. In this case, the two quantities can describe the same phenomenon at different time of the day, and therefore they can be combined; the first can be used to correct the second by a subtraction. The results of this analysis of correlation is presented in Figure 24 and Table 21.

The correlation analysis at Padova and Asiago-Ekar brings to values of the Pearson correlation coefficient equal to 0.75 and 0.93, respectively. The values of the slope are 1.24 and 1.14, confirming a possible small overestimate of the correction suggested by the twilight method.

The small differences from the unit value found for the sites of Padova and Asiago-Ekar can be justified by the different downward paths that the scattered light follows to reach the instrument when it comes from the Sun at twilight or when it is produced by artificial polluting sources on the ground.

For Passo Valles, the null correlation suggested by the twilight method causes the correlation to be zero between the brightness at twilight and the modal values of NSB as well.

For La Silla, a correlation is present, but weaker (0.64), and the value of the slope of the regression line is 0.6 ± 0.08, showing that the two quantities could be measured in different conditions.

For the sites of Passo Valles and La Silla, it has to be considered that the variations in both quantities are very small; they may not be large enough compared to other fluctuations in the environmental conditions, which could happen going from dusk to night and from night to dawn; it may be possible for these sites that are particularly open and have no natural obstacles nearby.

## 5. Conclusions

A significant decay of the spectral responsivity of the SQMs exposed to outdoor conditions was found. It could depend on the altitude of the site, and therefore on the radiation exposure, in particular to UV and short-wavelength radiations. The spectral responsivity of two aged SQMs was analysed; they operated outdoor at the same latitude, but at different altitudes: 12 m AMSL at Padova and 1050 m AMSL at Asiago-Pennar. An important degree of ageing was detected: the spectral responsivity at the wavelength of 450 nm was reduced to about 87% at Padova and to 60% for the SQM at Pennar, compared to the value of a new SQM. Even if the variation was significant, it did not completely explain the long-term trend highlighted by using the twilight method. In fact, considering the spectrum of the light scattered by the night sky, which would result in the worst case, i.e., choosing the larger deviation from the response of a new SQM, the loss of output was about 10 mmag_SQM_ arcsec^−2^ year^−1^ for the SQM of the Padova station and less than 40 mmag_SQM_ arcsec^−2^ year^−1^ for the SQM at Pennar. These values have to be compared with the trend found by the twilight method of 53 mmag_SQM_ arcsec^−2^ year^−1^ and 86 mmag_SQM_ arcsec^−2^ year^−1^ for the two sites, respectively. A significant difference was evident.

Variations in the atmospheric conditions could explain a fraction, even a significant one, of the residual change in the SQM outputs. These variations were analysed by observing the changes in the solar irradiance at ground level, as it is considered proportional to air transparency, and in the AOD values obtained from satellite measurements.

All the analysed stations indicated a decadal raise of the solar irradiance values, starting from the year 2014. It corresponded to a contemporaneous measured decrease in AOD, i.e., to an improvement of the air transparency over the analysed region. For the analysed Chilean site of La Silla, the solar irradiance measurements were not available, and only AOD was analysed. No significant long-term trend of AOD was detected, which means that there were no important variations in the atmosphere there.

Significant correlations were found between the values of the solar irradiance at ground level and the sky brightness at twilight and the monthly modal values of NSB; similar correlations were also found between AOD and the above cited SQM measurements. It confirms that the solar irradiance and AOD are different, but both effective, indicators of the status of the cleanliness of the atmosphere and of the evolution of the aerosol layer over our instruments used to track changes in NSB and light pollution.

An effect of variations in the air transparency was found different for different sites, depending on their locations. This was true of both the sky brightness at twilight, used for the SQM calibration, and for NSB.

The results underline again that the twilight calibration also accounts for atmospheric variations; therefore, the consequent correction of NSB depends on air transparency, and the entity of the dependence may vary with the site. Therefore, it cannot provide unambiguous information on the degradation of SQMs, which must be evaluated separately, for example, as presented in this work. On the contrary, precisely because the twilight method includes the atmospheric variations, this method makes it possible to compensate for them and to consider the evolution of NSB independent of changes in the composition and transparency of the air.

The analysis of solar irradiance or AOD versus sky brightness at twilight shows that the apparent brightness measured by the instruments at sites near or within light-polluting areas increases as the air transparency decreases. In our cases of Padova and Asiago-Ekar, but probably in general, these sites are below or within the main aerosol layer, which mainly scatters artificial light pollution downward. To compensate for the variations in the instrument output due to changes in the atmospheric conditions, the twilight method provides a data correction that is stronger the more the air conditions deviate from those on the NSB recording start date. For sites far from light-polluting sources, or more protected from them, the changes in the sky brightness at twilight are of minor amounts. For the Italian site of Passo Valles, a weak possible correlation was found between the solar irradiance and the sky brightness, while no significant correlation was found between AOD and brightness. The difference could be justified by remembering that, for this case, the solar irradiance values were recorded at the same site as the sky brightness, while the values of AOD were an average over almost the whole surface of the Veneto Region. The correlation with AOD at La Silla, a site far from polluting sources, was more significant but reflected mainly seasonal variations in AOD, and no significant long-term trend was found in either AOD or sky brightness at twilight.

The monthly modal values of NSB measured by the SQMs were found to be clearly correlated to the atmospheric conditions, analysing both the daily irradiance and AOD. For site close to light pollution sources, increasing air transparency leads to a darker sky, as the upward-directed light pollution is not scattered downward as much by air aerosol. In contrast, for sites far from light pollution sources, increasing air transparency leads to brighter skies, as more natural light can reach the instrument.

## Figures and Tables

**Figure 1 sensors-25-00516-f001:**
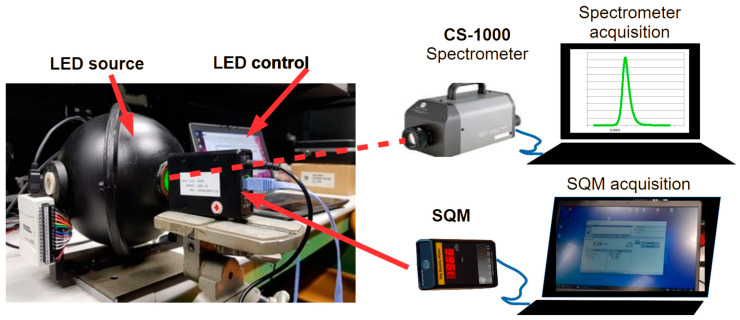
The test rig used to analyse the spectral responsivity of the SQMs; it is composed of an LED sphere, a spectroradiometer, and the SQM being tested.

**Figure 2 sensors-25-00516-f002:**
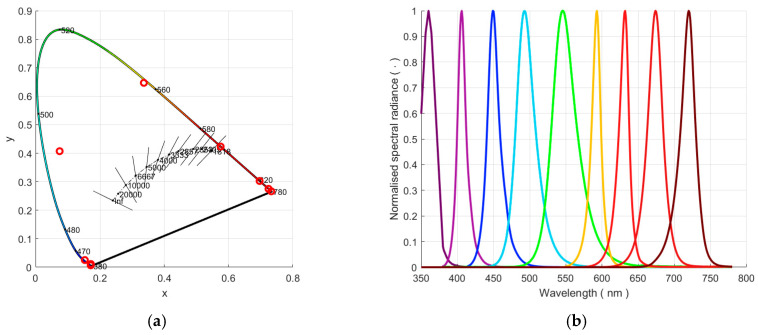
The x, y coordinates of the LED light in the CIE Lxy colour space (**a**); the spectral distributions of the light produced by each LED on the diffuser in the middle of the sphere (**b**).

**Figure 3 sensors-25-00516-f003:**
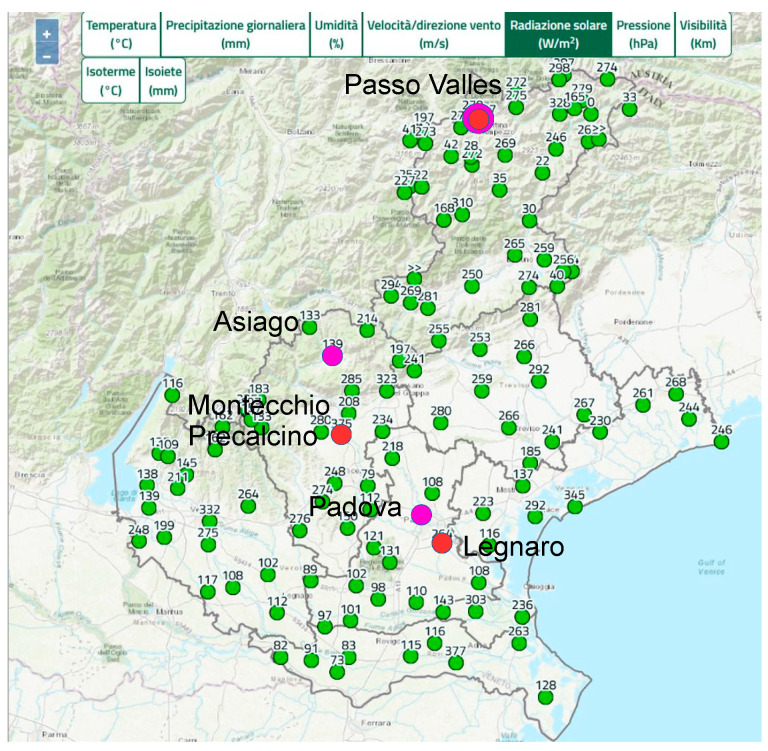
Maps of the sites of the ARPAV meteorological stations in the Veneto Region. The magenta dots represent the sites of the analysed SQMs (Padova, Asiago, and Passo Valles). The red dots show the sites close to them where the solar irradiance values were measured.

**Figure 4 sensors-25-00516-f004:**
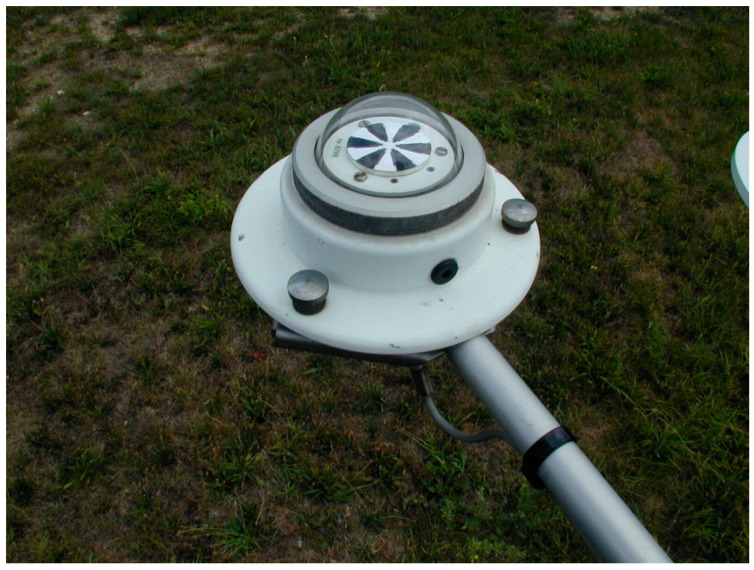
The global solar radiation sensor of the ARPAV meteorological stations, model SCHENK 8102.

**Figure 5 sensors-25-00516-f005:**
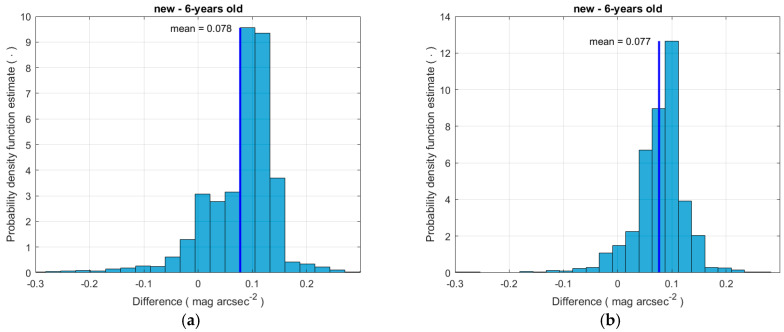
Histograms of SQM difference between measurements from the two different new SQMs and a 6-year-old SQM under the sky at Padova. The sections (**a**,**b**) refer to the two new instruments.

**Figure 6 sensors-25-00516-f006:**
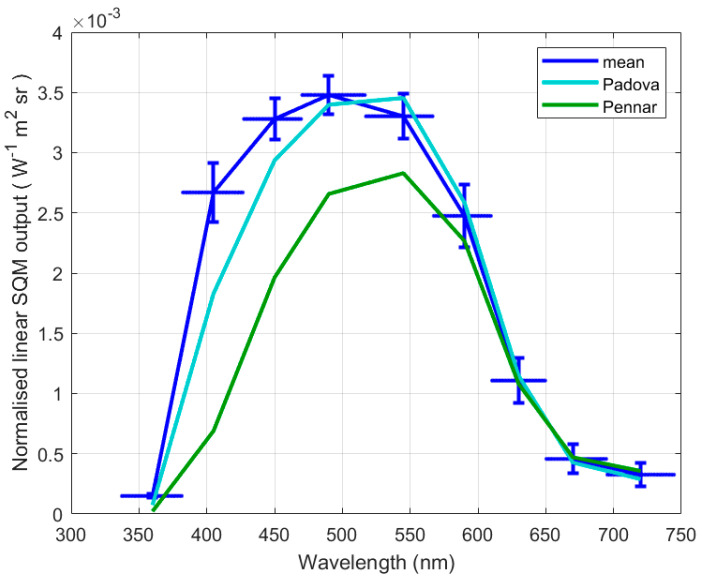
Output of the tested SQMs normalised to the sphere output for the 9 LEDs; the blue line corresponds to the average of 4 new SQMs and the one SQM often used as reference in outdoor comparison; the light blue line corresponds to the SQM at Padova after 6 working years; the green line corresponds to the SQM at Asiago-Pennar after 10 working years. The vertical bars represent the standard deviation among the five new instruments, while the horizontal bar represents the LED bandwidth.

**Figure 7 sensors-25-00516-f007:**
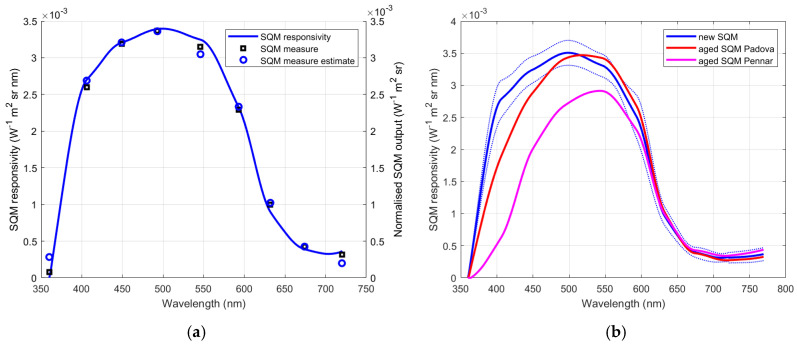
Estimated spectral responsivity of the analysed SQMs. (**a**) The continuous line represent the average responsivity of the new SQM; the black squares represent the normalised SQM measures, as in Figure 6; and the blue circles show the estimates of the measures calculated from the spectral responsivity. (**b**) The continuous blue line shows the average spectral responsivity of new SQMs with its uncertainty (blue dotted lines), the red line corresponds to the SQM used at Padova after 6 years, and the magenta line represents the SQM used 10 years at Asiago-Pennar.

**Figure 8 sensors-25-00516-f008:**
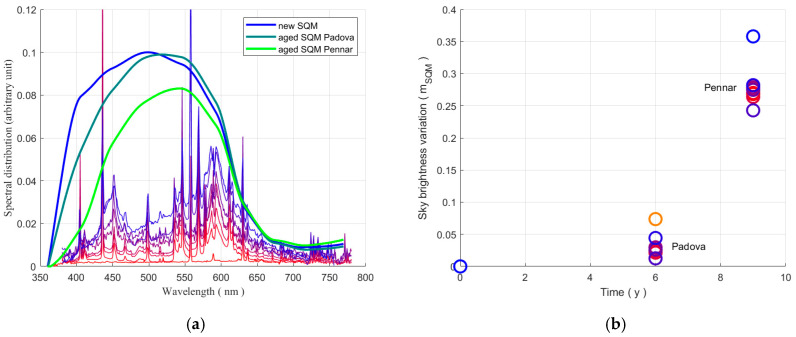
Spectral responsivity of new and aged SQMs and spectra of the night sky, the spectra with lower values correspond to low light polluted skies, represented by the red line, the higher one to the most light-polluted sky, represented by the blue line (**a**). Output variations of the two aged SQMs as responses of the different sky spectra, compared to a new SQM (**b**).

**Figure 9 sensors-25-00516-f009:**
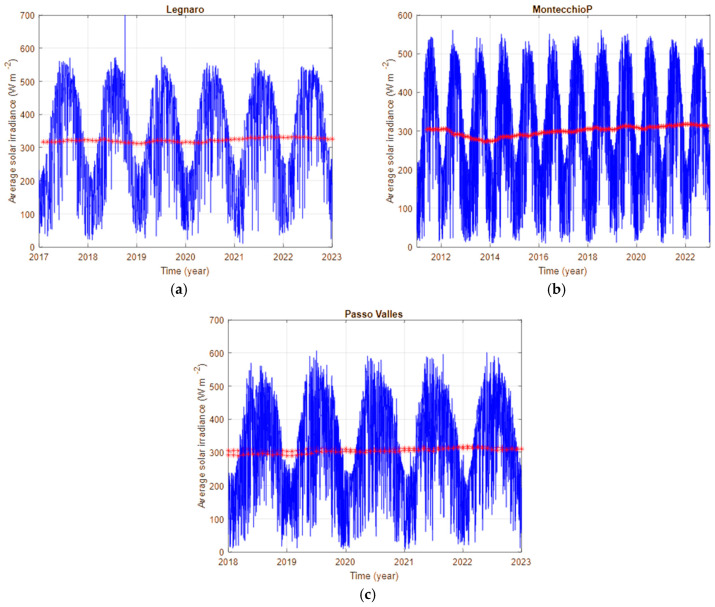
Average daily solar irradiance at the selected sites: (**a**) Legnaro, (**b**) Montecchio Precalcino, and (**c**) Passo Valles. The blue lines show the daily values, and the red lines represent the result of a moving average action with a window width of 2 years.

**Figure 10 sensors-25-00516-f010:**
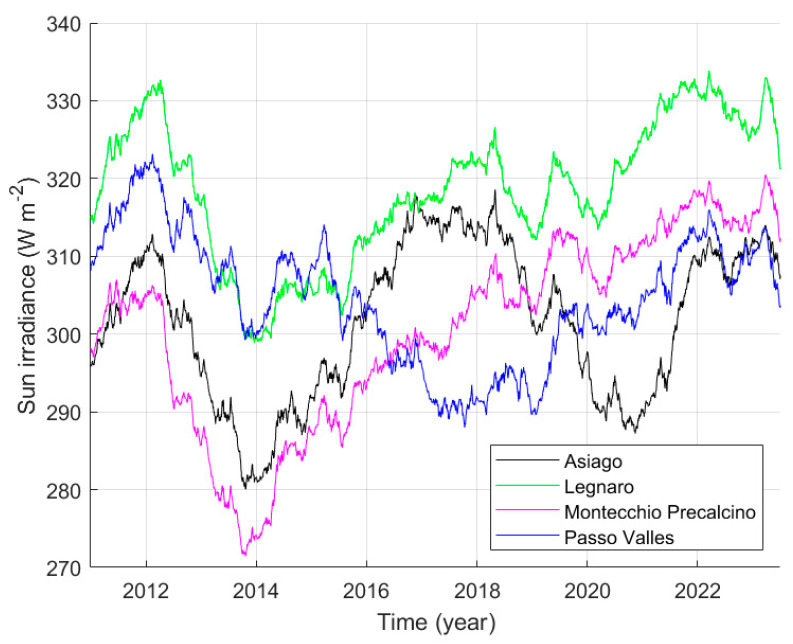
Solar irradiance averaged over a 2-year sliding window at Legnaro, Montecchio Precalcino, Passo Valles, and Asiago. Asiago is the station closest to Ekar but presents a significant effect of the presence of clouds.

**Figure 11 sensors-25-00516-f011:**
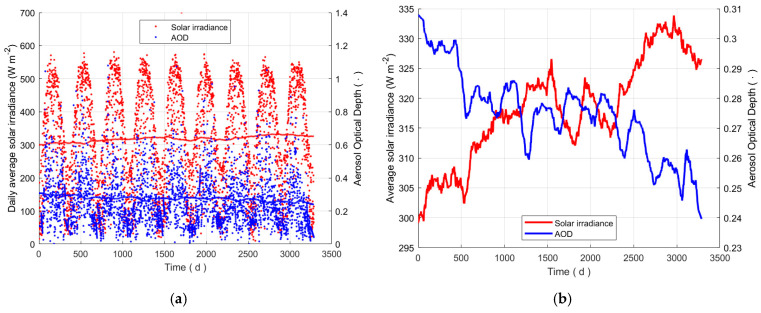
Comparison of the time evolution of the solar radiation (red dots and lines) and AOD (blue dots and lines) at Padova: daily data (**a**) and long-term trends obtained by averaging daily data using a 2-year sliding window (**b**).

**Figure 12 sensors-25-00516-f012:**
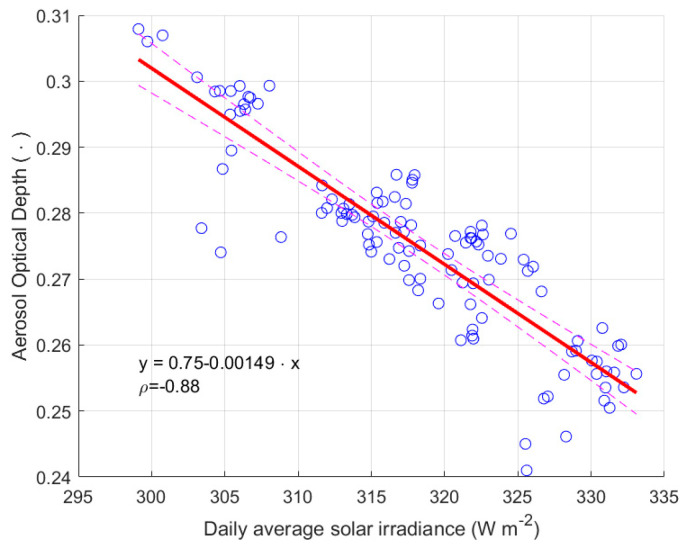
Correlation between monthly average solar radiation and AOD at Padova. The blue circles represent the monthly values, and the red line shows the result of the regression analysis, whose boundary with 95% uncertainty is shown by the dashed lines.

**Figure 13 sensors-25-00516-f013:**
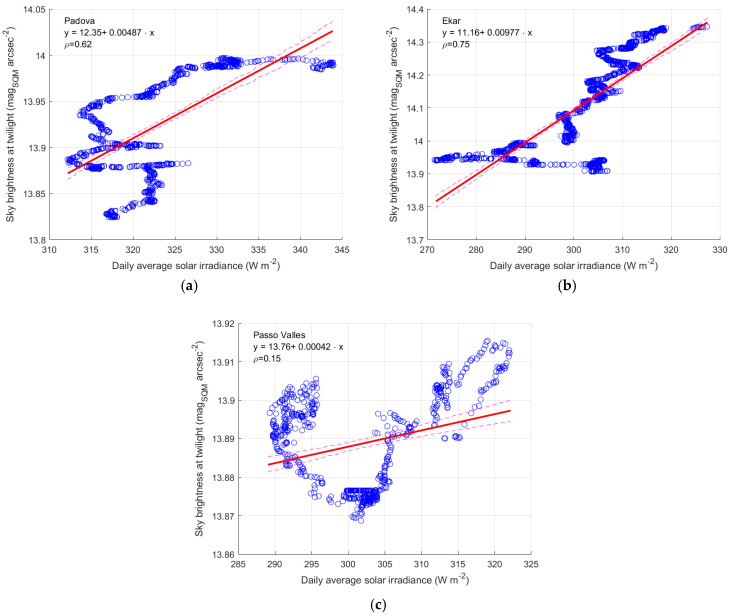
SQM measurements of the sky brightness at dusk versus daily solar irradiance for the three analysed Italian sites, Padova (**a**), Asiago-Ekar (**b**), and Passo Valles (**c**). A moving average with a 2-year sliding window is applied to daily data.

**Figure 14 sensors-25-00516-f014:**
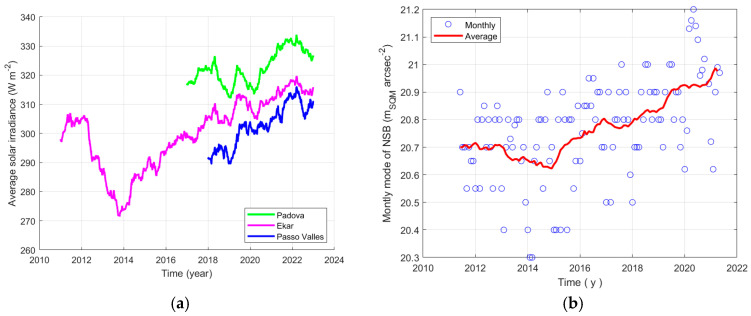
(**a**) Sun irradiance averaged over a 2-year sliding window at Legnaro, Montecchio Precalcino, Passo Valles; in the legend the names of the SQM sites are shown according to Table 1. The starting points of the curves correspond to the first data of the related SQM stations. (**b**) Circles represent the values of the monthly mode of the SQM measures at Asiago-Ekar, and the continuous line shows the average trend over a 2-year sliding window. It is an example of the available SQM data.

**Figure 15 sensors-25-00516-f015:**
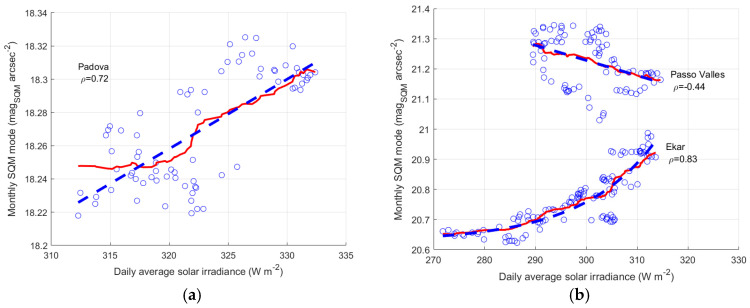
Monthly mode of NSB measured by the SQM versus the daily average solar irradiance at Padova (**a**) and at the site of Asiago-Ekar and Passo Valles (**b**). The average trends are shown by the red continuous lines. The blue curves show the behaviour of the fitting functions.

**Figure 16 sensors-25-00516-f016:**
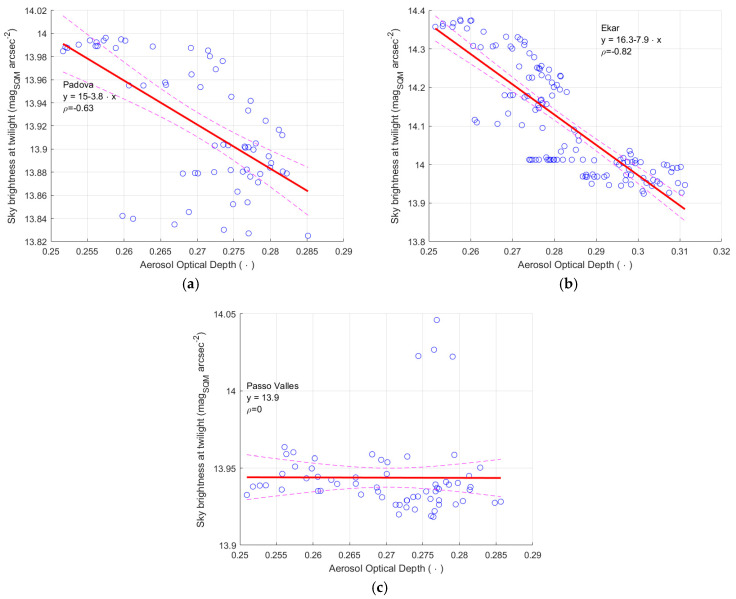
Relationship between aerosol optical depth and sky brightness at twilight at Padova (**a**), Ekar (**b**), and Passo Valles (**c**). Both AOD and sky brightness are values obtained from a moving average with a 2-year sliding window. Data are represented by blue circles, their linear regression is represented by the continuous red line, and the dashed lines show the boundary of the regression with a 95% confidence level.

**Figure 17 sensors-25-00516-f017:**
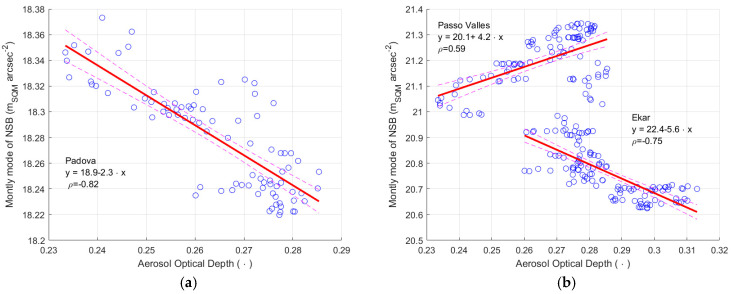
Relationship between the aerosol optical depth and the modal values of the night sky brightness at Padova (**a**) and at Ekar and Passo Valles (**b**). The values of both AOD and the NSB mode are obtained from a moving average with a 2-year sliding window. Monthly data are represented by blue circles, their linear regression is shown by the continuous red line, and the dashed lines indicate the boundary of the regression with a 95% confidence level.

**Figure 18 sensors-25-00516-f018:**
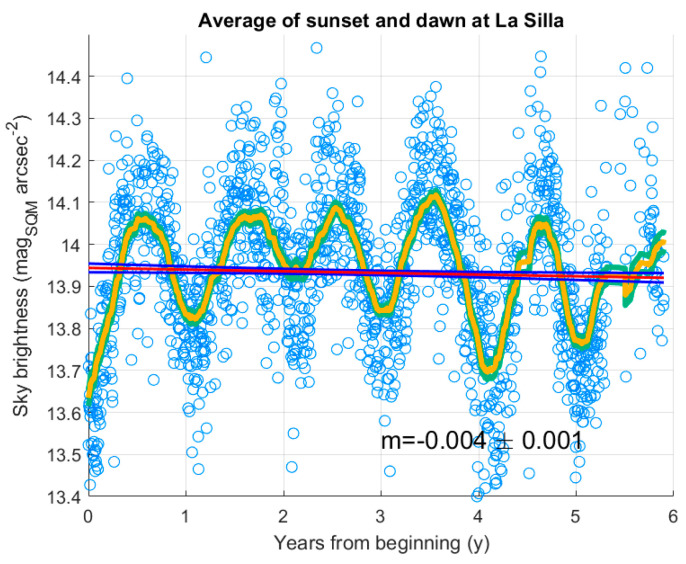
Average sky brightness at sunset and dawn at La Silla; circles are daily values; the yellow curve represents the averaged value in a sliding window 4 months wide; the green curves represent the variability of the average, accounting for one standard deviation; the red line is the linear regression, with its boundary in blue.

**Figure 19 sensors-25-00516-f019:**
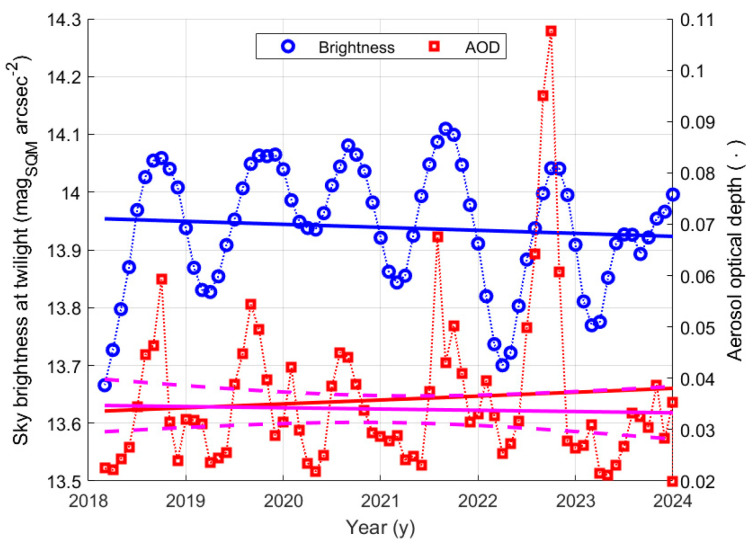
Monthly average of the values of the sky brightness at twilight (blue symbols and line) and of the aerosol optical depth at La Silla in the period from 2018 to the beginning of 2024.

**Figure 20 sensors-25-00516-f020:**
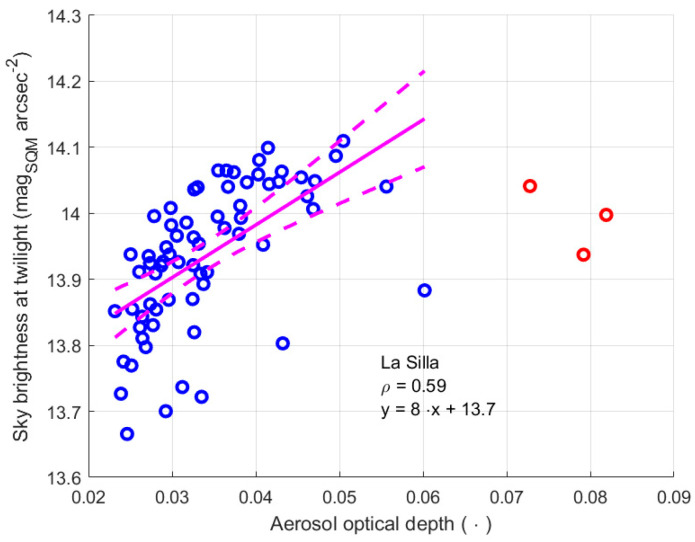
Sky brightness at twilight versus the aerosol optical depth at La Silla. The straight line represents the linear regression, and the dashed lines show the boundary with a 95% confidence level. The red circles correspond to data with AOD > 0.07, not included in the regression analysis.

**Figure 21 sensors-25-00516-f021:**
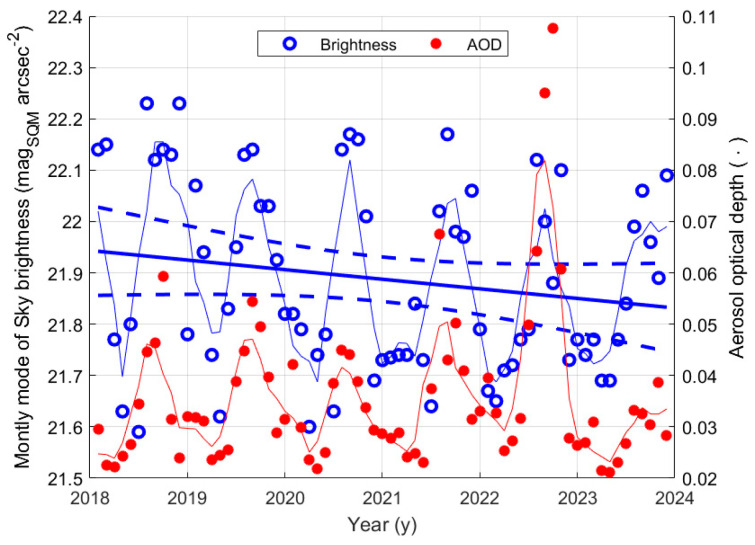
Monthly mode of the values the of the sky brightness at night (blue symbols and line) and of the aerosol optical depth at La Silla in the period from 2018 and the beginning of 2024.

**Figure 22 sensors-25-00516-f022:**
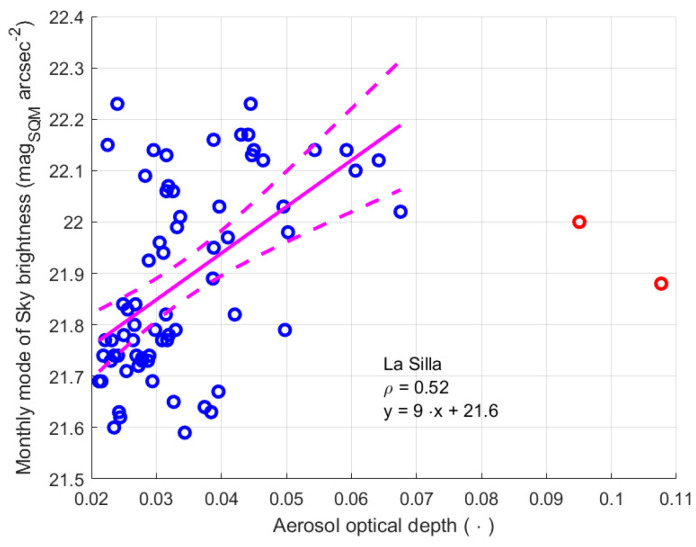
Values of the mode of the night sky brightness versus the aerosol optical depth at La Silla. The straight line represents the linear regression, and the dashed lines show the boundary with a 95% confidence level. The red circles correspond to data with AOD > 0.07, not included in the regression analysis.

**Figure 23 sensors-25-00516-f023:**
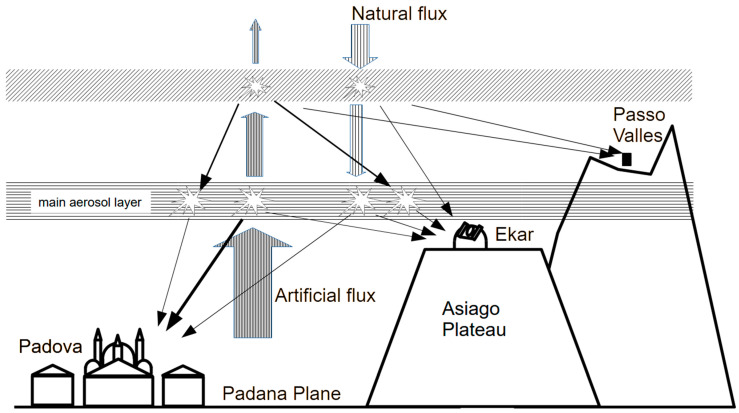
Sketch—not to scale—of the diffusion of the upward-directed artificial light and downward-directed natural light for sites with different positions relative to the main light pollution sources and aerosol layers. Wider or thicker arrows represent greater luminous flux involved.

**Figure 24 sensors-25-00516-f024:**
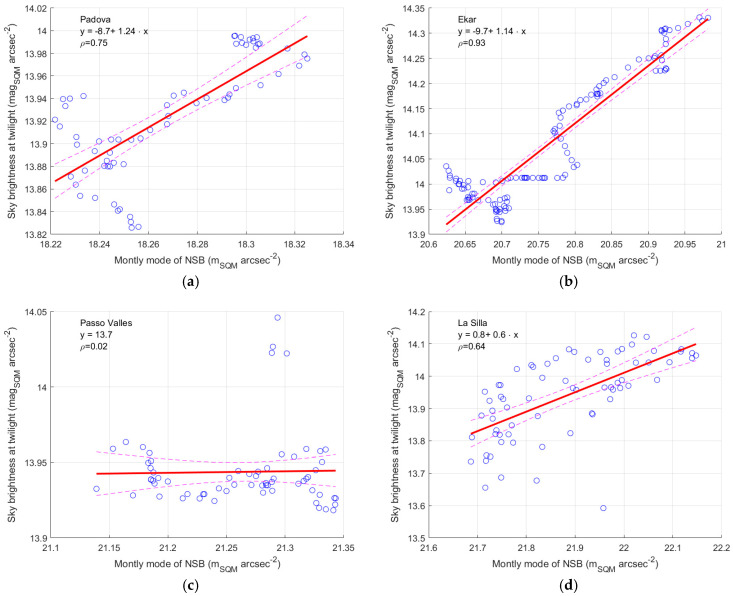
Correlation analysis of the brightness at twilight and the modal values of NSB for the four analysed sites: Padova (**a**), Asiago-Ekar (**b**), Passo Valles (**c**), and La Silla (**d**). The blue circles represent the monthly values, the solid red line shows the result of the linear regression, and the dashed lines describe the boundary of the regression with a 95% confidence level.

**Table 1 sensors-25-00516-t001:** Geographic coordinates (GPS coordinates and altitude above sea level) of the sites devoted to the measurements of NBS and of the solar irradiance.

Site	NSB	E_m_
Padova—Legnaro	45°24′23″ N 11°52′40″ E, h = 12 m	45°20′44″ N 11°57′50″ E, h = 8 m
Ekar—Montecchio Precalcino	45°50′55″ N 11°34′08″ E, h = 1366 m	45°39′57″ N 11°33′48″ E, h = 84 m
Passo Valles	45°20′19″ N 11°48′03″ E, h = 2032 m	

**Table 2 sensors-25-00516-t002:** Estimated coefficient of correlation and coefficients of the linear regression model y ~ p1 x+p2 for the averaged values of AOD versus the averaged values of the solar irradiance at Padova.

	Estimate	SD	Sample Size	*p* Value
**Pearson correlation coefficient**	−0.88	0.04	110	<<0.01
**Linear regression model**			110	
p1 (mag_SQM_ arcsec^−2^ W^−1^ m^2^)	−1.49 × 10^−3^	0.075 × 10^−3^		<<0.01
p2 (mag_SQM_ arcsec^−2^)	0.748	0.024		<<0.01

**Table 3 sensors-25-00516-t003:** Estimated coefficient of correlation and coefficients of the linear regression model y ~ p1 x+p2 for the sky brightness at twilight versus the daily average solar irradiance at Padova.

	Estimate	SD	Sample Size	*p* Value
**Pearson correlation coefficient**	0.62	0.04	871	<<0.01
**Linear regression model**			871	
p1 (mag_SQM_ arcsec^−2^ W^−1^ m^2^)	4.8·10^−3^	0.2·10^−3^		<<0.01
p2 (mag_SQM_ arcsec^−2^)	12.35	0.06		<<0.01

**Table 4 sensors-25-00516-t004:** Estimated coefficient of correlation and coefficients of the linear regression model y ~ p1 x+p2 for the sky brightness at twilight versus the daily average solar irradiance at Asiago-Ekar.

	Estimate	SD	Sample Size	*p* Value
**Pearson correlation coefficient**	0.747	0.024	1265	<<0.01
**Linear regression model**			1265	
p1 (mag_SQM_ arcsec^−2^ W^−1^ m^2^)	9.7·10^−3^	0.2·10^−3^		<<0.01
p2 (mag_SQM_ arcsec^−2^)	11.16	0.06		<<0.01

**Table 5 sensors-25-00516-t005:** Estimated coefficient of correlation and coefficients of the linear regression model y ~ p1 x+p2 for the sky brightness at twilight versus the daily average solar irradiance at Passo Valles.

	Estimate	SD	Sample Size	*p* Value
**Pearson correlation coefficient**	0.15	0.075	660	<<0.01
**Linear regression model**			660	
p1 (mag_SQM_ arcsec^−2^ W^−1^ m^2^)	0.42·10^−3^	0.05·10^−3^		<<0.01
p2 (mag_SQM_ arcsec^−2^)	13.76	0.015		<<0.01

**Table 6 sensors-25-00516-t006:** Estimated coefficient of correlation and coefficients of the linear regression model y ~ p1 x+p2 for the modal values of NSB versus the daily average solar irradiance at Padova.

	Estimate	SD	Sample Size	*p* Value
**Pearson’s correlation coefficient**	0.72	0.11	72	<<0.01
**Linear regression model**			72	
p1 (mag_SQM_ arcsec^−2^ W^−1^ m^2^)	4.2·10^−3^	4.8·10^−4^		<<0.01
p2 (mag_SQM_ arcsec^−2^)	16.9	0.16		<<0.01

**Table 7 sensors-25-00516-t007:** Estimated coefficient of correlation and coefficients of the linear regression model $y\;\sim \;p3+p2\;\cdot e^{\left[p1\cdot (x-mean(x_{meas}))\right]}$ for the modal values of NSB versus the daily average solar irradiance at Asiago-Ekar.

	Estimate	SD	Sample Size	*p* Value
**Pearson’s correlation coefficient**	0.83	0.06	119	<<0.01
**Exponential model**			119	
p1 (W^−1^ m^2^)	0.068	0.011		<<0.01
p2 (mag_SQM_ arcsec^−2^)	0.11	0.023		<<0.01
p3 (mag_SQM_ arcsec^−2^)	20.63	0.02		<<0.01

**Table 8 sensors-25-00516-t008:** Estimated coefficient of correlation and coefficients of the linear regression model y ~ p1 x+p2 for the modal values of NSB versus the daily average solar irradiance at Passo Valles.

	Estimate	SD	Sample Size	*p* Value
**Pearson’s correlation coefficient**	−0.44	0.17	85	<<0.01
**Linear regression model**			85	
p1 (mag_SQM_ arcsec^−2^ W^−1^ m^2^)	−5.0·10^−3^	1.1·10^−3^		<<0.01
p2 (mag_SQM_ arcsec^−2^)	22.7	0.34		<<0.01

**Table 9 sensors-25-00516-t009:** Estimated coefficient of correlation and coefficients of the linear regression model y ~ p1 x+p2 for the values of AOD versus the sky brightness at twilight at Padova; a moving average with a sliding window of 2 years was applied to both quantities.

	Estimate	SD	Sample Size	*p* Value
**Pearson’s correlation coefficient**	−0.63	0.15	64	<<0.01
**Linear regression model**			64	
p1 (mag_SQM_ arcsec^−2^)	−3.81	0.50		<<0.01
p2 (mag_SQM_ arcsec^−2^)	14.95	0.16		<<0.01

**Table 10 sensors-25-00516-t010:** Estimated coefficient of correlation and coefficients of the linear regression model y ~ p1 x+p2 for the values of AOD versus the sky brightness at twilight at Asiago-Ekar; a moving average with a sliding window of 2 years was applied to both quantities.

	Estimate	SD	Sample Size	*p* Value
**Pearson’s correlation coefficient**	−0.82	0.06	131	<<0.01
**Linear regression model**			131	
p1 (mag_SQM_ arcsec^−2^)	−7.9	0.5		<<0.01
p2 (mag_SQM_ arcsec^−2^)	16.33	0.13		<<0.01

**Table 11 sensors-25-00516-t011:** Estimated coefficient of correlation and coefficients of the linear regression model y ~ p1 x+p2 for the values of AOD versus the sky brightness at twilight at Passo Valles; a moving average with a sliding window of 2 years was applied to both quantities.

	Estimate	SD	Sample Size	*p* Value
**Pearson’s correlation coefficient**	0.00	0.24	65	0.997
**Linear regression model**			65	
p1 (mag_SQM_ arcsec^−2^)	−0.01	0.34		0.997
p2 (mag_SQM_ arcsec^−2^)	13.9	0.1		<<0.01

**Table 12 sensors-25-00516-t012:** Estimated coefficients of the linear regression model y ~ p1 x+p2 for the values of AOD versus the values of monthly mode of the night sky brightness at Padova; a moving average with a sliding window of 2 years was applied to both quantities.

	Estimate	SD	Sample Size	*p* Value
**Pearson’s correlation coefficient**	−0.85	0.07	85	<<0.01
**Linear regression model**			85	
p1 (mag_SQM_ arcsec^−2^)	−2.33	0.18		<<0.01
p2 (mag_SQM_ arcsec^−2^)	18.90	0.05		<<0.01

**Table 13 sensors-25-00516-t013:** Estimated coefficients of the linear regression model y ~ p1 x+p2 for the values of AOD versus the values of monthly mode of the night sky brightness at Ekar; a moving average with a sliding window of 2 years was applied to both quantities.

	Estimate	SD	Sample Size	*p* Value
**Pearson’s correlation coefficient**	−0.75	0.08	120	<<0.01
**Linear regression model**			120	
p1 (mag_SQM_ arcsec^−2^)	−5.6	0.5		<<0.01
p2 (mag_SQM_ arcsec^−2^)	22.37	0.13		<<0.01

**Table 14 sensors-25-00516-t014:** Estimated coefficients of the linear regression model y ~ p1 x+p2 for the values of AOD versus the values of monthly mode of the night sky brightness at Passo Valles; a moving average with a sliding window of 2 years was applied to both quantities.

	Estimate	SD	Sample Size	*p* Value
**Pearson’s correlation coefficient**	0.59	0.13	99	<<0.01
**Linear regression model**			99	
p1 (mag_SQM_ arcsec^−2^)	4.2	0.6		<<0.01
p2 (mag_SQM_ arcsec^−2^)	20.07	0.15		<<0.01

**Table 15 sensors-25-00516-t015:** Estimated coefficients of the time linear regression model y ~ p1 x + p2 for the brightness at La Silla at twilight over 6 years.

	Estimate	SD	Sample Size	*p* Value
p1 (mag_SQM_ arcsec^−2^ y^−1^)	−0.0005	0.007	71	0.49
p2 (mag_SQM_ arcsec^−2^)	24.5	20		0.10

**Table 16 sensors-25-00516-t016:** Estimated coefficients of the linear regression model y ~ p1 x+p2 for the AOD at La Silla over a 6-year time period.

	Estimate	SD	Sample Size	*p* Value
**AOD ≤ 0.07**			70	
p1 (y^−1^)	−0.00024	0.00075		0.75
p2 (·)	0.5	1.5		0.73
**All AOD data**			72	
p1 (y^−1^)	0.00075	0.0011		0.48
p2 (·)	−1.5	2.1		0.48

**Table 17 sensors-25-00516-t017:** Estimated coefficient of correlation and coefficients of the linear regression model y ~ p1 x+p2 for the values of AOD versus the sky brightness at twilight at La Silla; a moving average with a sliding window of 2 years was applied to both quantities.

	Estimate	SD	Sample Size	*p* Value
**Pearson’s correlation coefficient**	0.59	0.16	69	<<0.01
**Linear regression model**			69	
p1 (mag_SQM_ arcsec^−2^)	8.0	2.7		<<0.01
p2 (mag_SQM_ arcsec^−2^)	13.7	0.1		<<0.01

**Table 18 sensors-25-00516-t018:** Estimated coefficients of the linear regression model y ~ p1 x+p2 for the mode values of the night sky brightness at La Silla over 6 years of recording.

	Estimate	SD	Sample Size	*p* Value
p1 (mag_SQM_ arcsec^−2^ y^−1^)	−0.019	0.013	71	0.02
p2 (mag_SQM_ arcsec^−2^)	59	26		0.15

**Table 19 sensors-25-00516-t019:** Estimated coefficient of correlation and coefficients of the linear regression model y ~ p1 x+p2 on the values of AOD versus the modal values of NSB at La Silla; a moving average with a sliding window of 2 years was applied to both quantities.

	Estimate	SD	Sample Size	*p* Value
**Pearson’s correlation coefficient**	0.52	0.16	69	<<0.01
**Linear regression model**			69	
p1 (mag_SQM_ arcsec^−2^)	9.0	3.6		<<0.01
p2 (mag_SQM_ arcsec^−2^)	21.6	0.1		<<0.01

**Table 20 sensors-25-00516-t020:** Values of the coefficients of determination for the four analysed sites and the different analyses of the influence of air transparency.

	E_m_ vs. B_dusk_	AOD vs. B_dusk_	E_m_ vs. M_NSB_	AOD vs. M_NSB_
**Padova**	**0.75**	**0.40**	**0.52**	**0.67**
**Ekar**	**0.78**	**0.68**	**0.78**	**0.56**
**Passo Valles**	0.06	<10^−4^	0.19	**0.35**
**La Silla**	-	**0.35**	-	**0.27**

**Table 21 sensors-25-00516-t021:** Estimated coefficients of the linear regression model y ~ p1 x+p2 on the values of the brightness at twilight and the modal values of NSB for the four analysed sites.

	Estimate	SD	Sample Size	*p* Value
**Padova**				
**Pearson’s correlation coefficient**	0.75	0.11	64	<<0.1
**Linear regression model**			64	
p1 (·)	1.24	0.14		<<0.01
p2 (mag_SQM_ arcsec^−2^)	−8.7	2.5		<<0.1
**Asiago-Ekar**				
**Pearson’s correlation coefficient**	0.93	0.03	120	<<0.01
**Linear regression model**			120	
p1 (·)	1.14	0.04		<<0.01
p2 (mag_SQM_ arcsec^−2^)	−9.7	0.9		<<0.01
**Passo Valles**				
**Pearson’s correlation coefficient**	0.02	0.24	65	0.85
**Linear regression model**			65	
p1 (·)	0.01	0.05		0.85
p2 (mag_SQM_ arcsec^−2^)	13.7	1.1		<<0.01
**La Silla**				
**Pearson’s correlation coefficient**	0.64	0,14	70	<<0.01
**Linear regression model**			70	
p1 (·)	0.602	0.086		<<0.01
p2 (mag_SQM_ arcsec^−2^)	0.77	1.9		0.68

## Data Availability

Data underlying this paper will be shared on reasonable request to the corresponding author Pietro Fiorentin (pietro.fiorentin@unipd.it).

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
