# Peer review of "SQM Ageing and Atmospheric Conditions: How Do They Affect the Long-Term Trend of Night Sky Brightness Measurements?"

_sensors, 2025, doi:10.3390/s25020516_

Round 1

Reviewer 1 Report

Comments and Suggestions for Authors

Reviewer 2 Report

Comments and Suggestions for Authors

More details needed on the actual external setup of the SQM (glass dome for instance) and its possible contribution.

SQM's are notorious for low second digit accuracy. This should be mentioned and taken into account at some point.

Old SQM's performace as new is unknown. Is it identical to new ones? Has the industrial process of constructing them remained completely intact over the years?

Round 2

Reviewer 1 Report

Comments and Suggestions for Authors

Dear Authors, I am fine with your replies, I have only a doubt about the sign in the formula of AOD at line 295, Please check it.

Congratulations for your work

Author Response

Thank you for the report, actually there was typo.

The numerator and denominator have been swapped, according to the formula used in [27].